On the relationship between the mesospheric sodium layer and the meteoric input function
**Yanlin Li[1], Tai-Yin Huang[2], Julio Urbina[1], Fabio Vargas[3], Wuhu Feng[4]**
1.   Department of Electrical Engineering, Pennsylvania State University, University Park, PA, USA
2.   Department of Physics, Penn State Lehigh Valley, Center Valley, PA, USA
3.   Department of Electrical Engineering, University of Illinois Urbana-Champaign, Champaign, IL, USA
4.   National Centre for Atmospheric Science, University of Leeds, Leeds, UK
*Correspondence to*: Tai-Yin Huang (tuh4@psu.edu)
**Abstract**
This study examines the relationship between the concentration of atmospheric sodium and its
Meteoric Input Function (MIF). We use the measurements from the Colorado State University (CSU)
Lidar and the Andes Lidar Observatory (ALO) with a new numerical model that includes sodium
chemistry in the mesosphere and lower thermosphere (MLT) region. The model is based on the
continuity equation to treat all sodium-bearing species and runs at a high temporal resolution. The
model simulation employs data assimilation to compare the MIF inferred from the meteor radiant
distribution and the MIF derived from the new sodium chemistry model. The simulation captures the
seasonal variability of sodium number density compared with lidar observations over the CSU site.
However, there were discrepancies for the ALO site, which is close to the South Atlantic Anomaly (SAA)
region, indicating it is challenging for the model to capture the observed sodium over ALO. The CSU site
had significantly more lidar observations (27,930 hours) than the ALO sites (1872 hours). The simulation
revealed that the uptake of the sodium species on meteoric smoke particles was a critical factor in
determining the sodium concentration in MLT, with the sodium removal rate by uptake found to be
approximately three times that of the $NaHCO_3$ dimerization. Overall, the study's findings provide
valuable information on the correlation between MIF and sodium concentration in the MLT region,
contributing to a better understanding of the complex dynamics in this region. This knowledge can
inform future research and guide the development of more accurate models to enhance our
comprehension of the MLT region's behavior.

**Keywords:** Sodium layer, sodium chemistry, meteor radiant distribution, meteoric input function

**Key points:**
- A high-time resolution, time-dependent Na chemistry model is developed.
- Ablated global meteoroid material inputs inferred from ALO and CSU observations are about
$83\pm28$ t d$^{-1}$ and $53\pm23$ t d$^{-1}$, respectively.
- The frequency of meteor occurrences might not provide a precise reflection of the mass of
meteoroid material input.


## 1.  Introduction

Micro-meteoroids enter the Earth's atmosphere day and night, depositing their constituents into the atmosphere via ablation, creating a region that hosts various metal species, for example, Fe, K, Si, Mg, Ca, and Na, in both neutral and ion form (Plane et al., 2015; Plane et al., 2021; and references therein). The region is commonly referred to as the mesosphere and lower thermosphere (MLT), located between 75 and 110 km altitude. The metal layers in the MLT often serve as the tracers that facilitate the investigation of the dynamical and chemical processes within the region (Takahashi et al., 2014; Qiu et al., 2021). Quantitative measurements of metal atoms have been made since the 1950s (Hunten, 1967) through a variety of ground or space-borne technologies (Koch et al., 2021; Koch et al., 2022). The large resonant scattering cross-section (Bowman et al., 1969) and the substantial presence of the sodium atom in the MLT make it one of the most researched metal layers in the atmosphere (Yu et al., 2022).

The sodium layer is usually studied via observations carried out by resonance lidars, satellites, and through Na D-line emission at 589.0nm and 589.6nm (Plane, 2010; Plane et al., 2012; Hedin and Gumbel, 2011; Langowski et al., 2017; Andrioli et al., 2019; Li et al., 2020a). The sodium vertical profiles retrieved by lidars have been commonly used as a tracer to study atmospheric dynamics, e.g., gravity waves, wind shear, etc. The long-term seasonal and short-term diurnal variability of metallic species have been investigated by several studies (Feng et al., 2013; Marsh et al., 2013; Cai et al., 2019a, b; Yu et al., 2022; She et al., 2023). A typical sodium chemistry scheme consists of neutral chemistry, ion chemistry, and photolysis. The sodium chemistry research in recent years has primarily been based on the sodium chemistry model by Plane (2004), which has been cited in various subsequent works, including Bag et al. (2015) and references therein.

As meteoroids are the primary source of metal layers in the atmosphere, including the sodium layer, the Meteoric Input Function (MIF) plays a crucial role in the modeling of metallic layers in the atmosphere. The MIF is a function designed to model the temporal and spatial variability of the meteoroid on the atmosphere (Pifko et al., 2013). Sporadic meteors are estimated to make up more than 95% of the total meteoroid population by comparing the number of meteors originating in sporadic sources to those originating in known shower meteor sources (Chau and Galindo, 2008). This highlights the importance of incorporating sporadic meteor data in the MIF to accurately understand sodium concentration in the mesosphere and lower thermosphere (MLT) region and its correlation with meteoroid material input. It is well established that there are six apparent sources of sporadic meteors, namely North and South Apex (NA and SA); North and South Toroidal (NT and ST); and Helion and Anti-Helion sources (H and AH) (Campbell-Brown, 2008; Kero et al., 2012; Li et al., 2022). However, the relative strength of these meteor radiant sources varies among the studies. For example, the NA and SA sources are found to be much stronger than other sources in results obtained with High Power Large Aperture (HPLA) radars (Chau et al., 2007; Kero et al., 2012; Li and Zhou, 2019), while specular meteor radars found the difference to be much smaller (Campbell-Brown and Jones, 2005; Campbell-Brown, 2008). The detection sensitivity varies significantly among different facilities. For instance, the Arecibo Observatory (AO) at 18° N, 66° W detects approximately 20 times more meteors per unit area per unit time than the Jicamarca Radio Observatory (JRO) at 12° S, 77° W, and at least 800 times more meteors than the Resolute Bay Incoherent Scatter North (RISR-N) radar at 75° N, 95° W, despite all being HPLA facilities (Li et al., 2020, 2023a; Hedges et al., 2022). While meteor flux does exhibit variations based on time and latitude, these fluctuations alone cannot explain the magnitude of the observed difference.

Consequently, the total mass of the meteors that enter the Earth's atmosphere is subject to significant
uncertainties. In the existing Whole Atmosphere Community Climate Model-Na (WACCM-Na) global
sodium model (Dunker et al., 2015), the meteoric input function was modeled by placing a flux curve on
each radiant meteor source with a definite ratio (more details can be found in Marsh et al., 2013). The
flux curve model is based on observations carried out exclusively by the Arecibo Observatory. Although
the model can reproduce some of the flux characteristics of the meteors observed at Arecibo, it is a
relatively simple model and therefore has several limitations (Li et al., 2022). One of the limitations is
that the model cannot reproduce the velocity distribution of the meteors in observations.
This study introduces a new numerical model for sodium chemistry that utilizes the continuity equations
for all Na-related reactions without steady-state approximations. The main objective is to investigate the
relationship between the apparent sodium concentration and the MIF in the MLT region. We then
compare the results of the new model with measurements from two lidar instruments, namely the
Colorado State University (CSU) and the Andes Lidar Observatory (ALO). Furthermore, we compared the
MIF derived from the new sodium chemistry model and lidar measurements from CSU and ALO, against
the results of the high-resolution meteor radiant distribution recently deduced from observations
conducted at AO. Finally, we discuss the implications of these comparisons and suggest possible
explanations for the discrepancy between the MIF derived from radar and those obtained from lidar
observations.

**2. The sodium chemistry model (NaChem)**
**2.1 Sodium chemistry**
Numerical airglow models have been extensively used to investigate atmospheric airglow chemistry and
gravity waves (Huang and Hickey, 2008; Huang and Richard, 2014; Huang, 2015). A new numerical
sodium chemistry model, hereafter referred to as NaChem, was developed for this study. Table 1 lists
the complete reactions and their corresponding rate coefficients used in NaChem, which includes
neutral chemistry, ion chemistry, and photochemistry. The dimerization reaction of $NaHCO_3$ (reaction 25
in Table 1) is the outlet that removes Na atoms in the chemistry scheme. The Na atoms can also be
removed by the uptake of sodium species onto meteoric smoke particles (Hunten et al., 1980;
Kalashnikova et al., 2000; Plane, 2004), a process that can be turned on or off in the model. This study
estimates the MIF in the numerical model by matching the amount of sodium atoms removed by the
dimerization reaction and uptake, i.e., sodium sink, to maintain the observed sodium presence in the
MLT. MIF is a function of time and latitude, representing the mass of meteoroid material entering
Earth's atmosphere. Throughout the rest of the paper, the MIF estimated from the sodium chemistry
numerical model will be referred to as MIF(s). On the other hand, the MIF derived from meteor radiant
distribution, referred to as MIF(m). The MIF(m) is determined through a 3-D meteoroid orbital
simulation, a process similar to the seeding process discussed in section 3.1 of Li et al. (2022), based on
the meteor radiant distribution. MIF(m) is in arbitrary units. Note that the meteor mass cannot be
accurately determined via radar measurements, however, the seasonal variation of meteoroid material
input can be represented by MIF(m). The estimation of meteor mass is further discussed in Section 5. In
contrast, MIF(s) is expressed in units of $1/cm^3/second$.
The numerical model utilizes the continuity equation to track the time evolution of all 14 Na-related
species. Table 2 presents a comprehensive list of these species, along with their corresponding
production and loss rates. The background gas species, including $O_3$, $O_2$, O, H, $H_2$, $H_2O$, etc., and the
temperature are provided by WACCM version 6 (Jiao et al., 2022). Here we use the dynamic version of
WACCM nudged with NASA's Modern Era Retrospective Analysis for Research and Application MERRA2
reanalysis data set (Hunziker & Wendt, 1974; Molod et al., 2015; Gettelman et al., 2019). The WACCM
reference profiles are linearly interpolated to a resolution of one minute and updated every minute
during the simulation. It is worth noting that the Na-related reactions, which are illustrated in Table 2,
do not significantly impact the background gas species, as the effect is orders of magnitude smaller than
the variation of the major gas species themselves. Therefore, the major gas species are simulated
independently of Na-related reactions.
**2.2 Numerical scheme**
As discussed earlier, it is worth noting that the reactions of sodium chemistry in NaChem share
similarities with those in previous models (e.g., Plane et al., 2015 and references therein); however, the
implementation of the numerical chemistry scheme differs. NaChem uses continuity equations to treat
all chemicals involved, including short-lived intermediate species. Treating all species with the continuity
equation is a straightforward and more accurate approach than using steady-state approximations.
Moreover, by treating all species in a uniform procedure, the numerical model is more compact and
easier to interpret. The computational capability of a personal computer nowadays has advanced
enough to process an ultra-fine time step (microseconds) that is necessary for numerical simulations of
short-lived species in a reasonable duration. Still, the differential equations for production and loss of
short-lived species can be numerically unstable unless microsecond or even sub-microsecond time step
is used (Higham, 2002). The concern of the differential equation instability can be largely mitigated by a
first-order exponential integrator (Hochbruck and Ostermann, 2010), i.e.,
$$c = x_0 - \frac{a_0}{b_0} \qquad \text{(1a)}$$

$$x_1 = \frac{a_0}{b_0} + ce^{-b_0\Delta t} \quad \text{(1b)}$$

Where $x_0$ is the value of the current step. In the simulation, it is the number density of the species. $a_0$
$(1/cm^3/s)$ is the production of the species, $b_0$ $(1/s)$ is the loss rate of the species, $\Delta t$ is the step size in
time, and $x_1$ is the value of the next step. The units for $x_0$, $x_1$, and $c$ are $1/cm^3$.
The exponential integrator, as expressed in Eq. 1a and 1b, is the solution to the continuity equation.
Note that reaction 25 listed in Table 1 is an exception, which was carried out using explicit Euler
integrator in the simulation. This reaction's continuity equation is structured differently from the others
because it represents the only mechanism for removing Na atoms from the chemistry simulation, apart
from the uptakes of sodium species. Our testing indicates that both the exponential integrator and
explicit Euler integrator yield nearly identical results. However, for numerical stability, the explicit Euler
integrator requires a step size of ~1μs, which is orders of magnitude smaller than the exponential
integrator. The default time step of NaChem is 0.1 seconds with the exponential integrator.


Table 1. Reactions in NaChem. $f_a$ and $f_x$ are branching ratios.

| | Reaction | Rate Coefficient | reference |
|---|---|---|---|
| | **neutral chemistry** | | |
| 1 | $Na + O_3 \rightarrow NaO(A) + O_2$ | $K_1 = 1.1 \times 10^{-9} \exp(-116/T)$ | 1 |
| 2 | $NaO(A) + O \rightarrow Na(^2P_J) + O_2$ | $K_2 = 2.2 \times 10^{-10}(T/200)^{0.5}$, $f_A = 0.14 \pm 0.4$ | 1,3 |
| 3 | $NaO(A) + O \rightarrow Na(^2S) + O_2$ | $K_3 = 2.2 \times 10^{-10}(T/200)^{0.5}$, $(1-f_A)$ | 1,3 |
| 4 | $NaO(A) + O_2 \rightarrow NaO(X) + O_2$ | $K_4 = 1 \times 10^{-11}$ | 1 |
| 5 | $Na + O_2 + M \rightarrow NaO_2 + M$ | $K_5 = 5.0 \times 10^{-30}(200/T)^{1.22}$ | 1 |
| 6 | $NaO_2 + O \rightarrow NaO(X) + O_2$ | $K_6 = 5 \times 10^{-10} \exp(-940/T)$ | 1 |
| 7 | $NaO(X) + O \rightarrow Na(^2P_J) + O_2$ | $K_7 = 2.2 \times 10^{-10}(T/200)^{0.5}$, $f_x = 0.167$ | 1,2 |
| 8 | $NaO(X) + O \rightarrow Na(^2S) + O_2$ | $k_8 = 2.2 \times 10^{-10}(T/200)^{0.5}$, $(1-f_x)$ | 1,2 |
| 9 | $NaO(X) + O_3 \rightarrow NaO_2 + O_2$ | $k_9 = 1.1 \times 10^{-9} \exp(-568/T)$ | 1 |
| 10 | $NaO(X) + O_3 \rightarrow Na + 2O_2$ | $k_{10} = 3.2 \times 10^{-10} \exp(-550/T)$ | 1 |
| 11 | $NaO(X) + O_2 + M \rightarrow NaO_3 + M$ | $k_{11} = 5.3 \times 10^{-30}(200/T)$ | 1 |
| 12 | $NaO(X) + H \rightarrow Na + OH$ | $k_{12} = 4.4 \times 10^{-10} \exp(-668/T)$ | 1 |
| 13 | $NaO(X) + H_2 \rightarrow NaOH + H$ | $k_{13} = 1.1 \times 10^{-9} \exp(-1100/T)$ | 1 |
| 14 | $NaO(X) + H_2 \rightarrow Na + H_2O$ | $k_{14} = 1.1 \times 10^{-9} \exp(-1400/T)$ | 1 |
| 15 | $NaO(X) + H_2O \rightarrow NaOH + OH$ | $k_{15} = 4.4 \times 10^{-10} \exp(-507/T)$ | 1 |
| 16 | $NaO(X) + CO_2 + M \rightarrow NaCO_3 + M$ | $K_{16} = 1.3 \times 10^{-27}(200/T)$ | 1 |
| 17 | $NaO_2 + H \rightarrow Na + HO_2$ | $K_{17} = 1.0 \times 10^{-9} \exp(-1000/T)$ | 1 |
| 18 | $NaO_3 + O \rightarrow Na + 2O_2$ | $k_{18} = 2.5 \times 10^{-10}(T/200)^{0.5}$ | 1 |
| 19 | $NaCO_3 + O \rightarrow NaO_2 + CO_2$ | $k_{19} = 5.0 \times 10^{-10} \exp(-1200/T)$ | 1 |
| 20 | $NaCO_3 + H \rightarrow NaOH + CO_2$ | $k_{20} = 1.0 \times 10^{-9} \exp(-1400/T)$ | 1 |
| 21 | $NaOH + H \rightarrow Na + H_2O$ | $k_{21} = 4.0 \times 10^{-11} \exp(-550/T)$ | 1 |
| 22 | $NaOH + CO_2 + M \rightarrow NaHCO_3 + M$ | $k_{22} = 1.9 \times 10^{-28}(200/T)^1$ | 1 |
| 23 | $NaHCO_3 + H \rightarrow Na + H_2O + CO_2$ | $k_{23} = 1.1 \times 10^{-11} \exp(-910/T)$ | 1 |
| 24 | $NaHCO_3 + H \rightarrow Na + H_2CO_3$ | $k_{24} = 1.84 \times 10^{-13} T^{0.777} \exp(-1014/T)$ | 1 |
| 25 | $2NaHCO_3 + M \rightarrow (NaHCO_3)_2 + M$ | $k_{25} = 8.8 \times 10^{-10} \exp(T/200)^{-0.23}$ | 1 |
| 26 | $Na(^2P_J) \rightarrow Na(^2S) + h\nu(589.0-589.6\ nm)$ | $K_{26} = 6.26 \times 10^7$ | 1 |
| | **ion-molecule chemistry** | | |
| 27 | $Na + O_2^+ \rightarrow Na^+ + O_2$ | $K_{27} = 2.7 \times 10^{-9}$ | 1 |
| 28 | $Na + NO^+ \rightarrow Na^+ + NO$ | $K_{28} = 8.0 \times 10^{-10}$ | 1 |
| 29 | $Na^+ + N_2 + M \rightarrow NaN_2^+ + M$ | $k_{29} = 4.8 \times 10^{-30}(T/200)^{-2.2}$ | 1 |
| 30 | $Na^+ + CO_2 + M \rightarrow NaCO_2^+ + M$ | $k_{30} = 3.7 \times 10^{-29}(T/200)^{-2.9}$ | 1 |
| 31 | $NaN_2^+ + O \rightarrow NaO^+ + N_2$ | $k_{31} = 4.0 \times 10^{-10}$ | 1 |
| 32 | $NaO^+ + N_2 \rightarrow NaN_2^+ + O$ | $k_{32} = 1.0 \times 10^{-12}$ | 1 |
| 33 | $NaO^+ + O \rightarrow Na^+ + O_2$ | $k_{33} = 1.0 \times 10^{-11}$ | 1 |
| 34 | $NaO^+ + O_2 \rightarrow Na^+ + O_3$ | $k_{34} = 5.0 \times 10^{-12}$ | 1 |
| 35 | $NaN_2^+ + X \rightarrow NaX^+ + N_2\ (X=CO_2, H_2O)$ | $k_{35} = 6.0 \times 10^{-10}$ | 1 |
| 36 | $NaY^+ + e \rightarrow Na + Y\ (Y=N_2, CO_2, H_2O, O)$ | $k_{36} = 1.0 \times 10^{-6}(T/200)^{-0.5}$ | 1 |
| | **photochemical reactions** | | |
| 37 | $NaO(A)/NaO(X) + h\nu \rightarrow Na + O$ | $K_{37} = 5.5 \times 10^{-2}$ | 1 |
| 38 | $NaO2 + h\nu \rightarrow Na + O2$ | $K_{38} = 1.9 \times 10^{-2}$ | 1 |
| 39 | $NaOH + h\nu \rightarrow Na + OH$ | $K_{39} = 1.8 \times 10^{-2}$ | 1 |
| 40 | $NaHCO3 + h\nu \rightarrow Na + HCO3$ | $K_{40} = 1.3 \times 10^{-4}$ | 1 |
| 41 | $Na + h\nu \rightarrow Na+ + e-$ | $K_{41} = 2 \times 10^{-5}$ | 1 |

*1:Plane (2004), 2: Plane (2012), 3: Griffin et al. (2001). Units for rate coefficient: unimolecular, $s^{-1}$;
bimolecular, $cm^3\ s^{-1}$, termolecular, $cm^6 s^{-1}$, etc.
Table 2. The production and loss terms of the sodium-related species.

| | Species | Prod | Loss |
|---|---|---|---|
| a1 | $Na(^2P_J)$ | $k_2[a_3][O] + k_7[a_5][O];$ | 1* |
| a2 | Na | $k_3[a_3][O] + k_8[a_5][O] + k_{10}[a_5][O_3] + k_{12}[a_5][H] + k_{14}[a_5][H_2] + k_{17}[a_4][H] + k_{18}[a_6][O] + k_{21}[a_7][H] + k_{23}[a_9][H] + k_{24}[a_9][H] + k_{36}[a_{11}][e] + k_{36}[a_{13}][e] + k_{36}[a_{12}][e] + k_{36}[a_{14}][e] + [a_1] + k_{37}[a_3][hv] + k_{37}[a_5][hv] + k_{38}[hv][a_4] + k_{39}[hv][a_7] + k_{40}[hv][a_9];$ | $k_1[O_3] + k_5[O_3] + k_5[O_2][M] + k_{27}[O_2^+] + k_{28}[NO^+] + k_{41}[hv];$ |
| a3 | NaO(A) | $k_1[a_2][O_3]$ | $k_2[O] + k_3[O] + k_4[O_2] + k_{37}[hv]$ |
| a4 | $NaO_2$ | $k_5[a_2][O_2][M] + k_9[a_5][O_3] + k_{19}[a_8][O]$ | $k_6[O] + k_{17}[H] + k_{38}[hv]$ |
| a5 | NaO(X) | $k_5[a][O_3]+ k_4[a_3][O_2] + k_6[a_4][O]$ | $k_7[O] + k_8[O] + k_9[O_3] + k_{10}[O_3] + k_{11}[O_2][M] + k_{12}[H] + k_{13}[H_2] + k_{14}[H_2] + k_{15}[H_2O] + k_{16}[CO_2][M] + k_{37}[hv]$ |
| a6 | $NaO_3$ | $k_{11}[a_5][O_2][M]$ | $k_{18}[O]$ |
| a7 | NaOH | $k_{13}[a_5][H_2] + k_{15}[a_5][H_2O] + k_{20}[a_8][H]$ | $k_{21}[H] + k_{22}[CO_2][M] + k_{39}[hv]$ |
| a8 | $NaCO_3$ | $k_{16}[a_5][CO][M]$ | $k_{19}[O] + k_{20}[H]$ |
| a9 | $NaHCO_3$ | $k_{22}[a_7][CO_2][M]$ | $k_{23}[H] + k_{24}[H] + 2k_{25}[a_9][M] + k_{40}[hv]$ |
| a10 | $Na^+$ | $k_{27}[a_2][O_2^+] + k_{28}[a_2][NO^+] + k_{33}[a_{13}][O] + k_{34}[a_{13}][O_2] + k_{41}[hv][a_2]$ | $k_{29}[N_2][M] + k_{30}[CO_2][M]$ |
| a11 | $NaN_2^+$ | $k_{29}[a_{10}][N_2][M] + k_{32}[a_{13}][N_2]$ | $k_{31}[O]+k_{35}[CO_2] + k_{35}[H_2O] + k_{36}[e]$ |
| a12 | $NaCO_2^+$ | $k_{30}[a_{10}][CO_2][M] + k_{35}[a_{11}][CO_2]$ | $k_{36}[e]$ |
| a13 | $NaO^+$ | $k_{31}[a_{11}][O]$ | $k_{32}[N2]+k_{33}[O]+k_{34}[O_2]+k_{36}[e]$ |
| a14 | $NaH_2O^+$ | $k_{35}[a_{11}][H_2O]$ | $k_{36}[e]$ |


*In Species 1, as of the current state of the model, all $Na(^2P_J)$ atoms return to their ground state
immediately, so the loss term is set to 1. The [hv] is the term that represents loss via photoionization,
which is approximately a sinusoidal function based on the solar zenith angle of the respective local solar
time.

**3. CSU and ALO Sodium Lidar Observations and data processing**
**3.1 Observations**
Several aspects of the current research, i.e., the presence of sodium in the MLT, require cross-validation
with the measurements. One primary objective of the present model is to match the observed seasonal
variation of the sodium layer. Measurements by the Colorado State University (CSU, 41.4°N, 111.5°W)
Lidar, also known as Utah State University (USU) Lidar, and the lidar data acquired by the Andes Lidar
Observatory (ALO, 30.3°S, 70.7°W), are used to facilitate the research in the current study. We are
unable to acquire more ALO data after 2019 as the COVID situation disrupted the site operation. The
CSU data comprises 27,930 hours of lidar observations between 1990 and 2020, whereas the ALO data
consists of 1872 hours between 2014 and 2019.

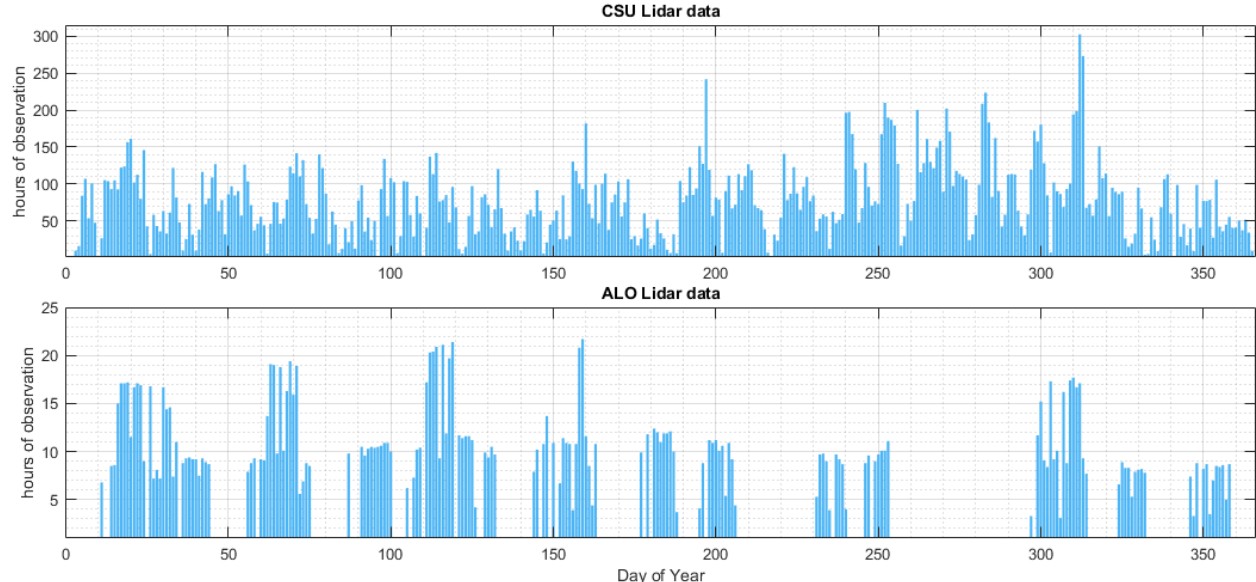


**Figure 1**. Available hours of lidar observations. CSU lidar (1990-2020, upper plot) and ALO lidar (2014-
2019, lower plot).
The statistics of CSU and ALO available data are presented in Figure 1. The Lidar observations of both
sites consist of nocturnal observations only, and a typical nocturnal observation lasts between 8 and 11
hours. Note that in Figure 1, there could be as many as 300 hours of sodium observations on a single day
of year, which means the data of the date comprise observations of many years on that day in different
years. The CSU data almost covered every day of the year with only a few exceptions, whereas the ALO
data was much more sparse. As a result, due to the significantly larger number of CSU observations, the
statistical reliability of the seasonal variation in the sodium layer derived from ALO observations may not
be as strong as that of the CSU data. As depicted in Figure 2, the overall seasonal trend of the sodium
vertical profile derived from CSU lidar observations closely aligns with the simulation-based estimate by
Marsh et al. (2013). In contrast, ALO lidar observations deviate from the findings reported by Marsh et
al. (2013). The ALO measurements exhibit a prominent peak around June, while the results in Marsh et
al. (2013) show a double peak in March and October.
**3.2 Data processing**
The sodium layer in atmospheric observations is often affected by perturbations of atmospheric
dynamics, which is why sodium is commonly used as a tracer in the study of the MLT dynamics (Plane et
al., 2015). However, studying the sodium layer itself can be complicated due to the underlying chemical
processes coupled with the dynamics. In order to mitigate the effects of atmospheric dynamics, we
process the sodium vertical profiles from observations in three steps. First, we average the profiles by
day of the year, meaning we take the average of the data from the same day of the year from different
years. Missing data are treated using linear interpolation. Next, we smooth the averaged profiles using a
15-day running average. Finally, the height profile for each time step is further smoothed by fitting it
with a skew-normal distribution (Azzalini & Valle, 1996), using the least squares error method.

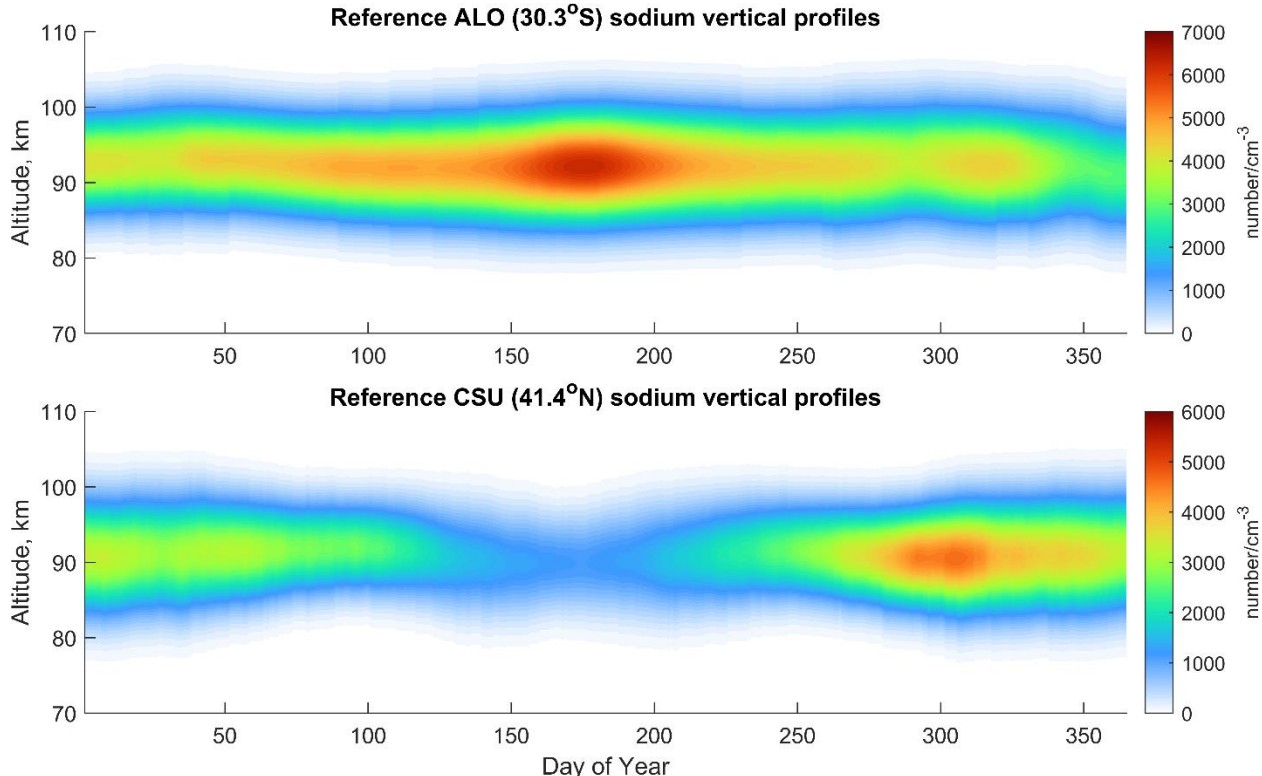


**Figure 2**. *The reference annual sodium vertical profiles at ALO (top plot) and at CSU (bottom plot). The*
*reference profiles are the averages throughout all the available data on the same days at the respective*
*site, then fitted by a skew-normal distribution that mitigates atmospheric dynamics. In essence, the*
*reference profiles are measurements with small-scale dynamics removed via steps discussed in section*
*3.2.*


Figure 2 displays the processed annual sodium vertical profiles from the lidar measurements, referred to
as reference profiles hereafter. These profiles serve as references to guide the numerical simulation of
the NaChem model. The reference profiles are Na lidar measurements fitted using a skew-normal
distribution, smoothed by a 15-day running average, and processed through linear 2-D interpolation
across time and altitude. The lidar measurements have an altitude resolution of 500m for ALO and from
75m to 140m for CSU. These measurements are interpolated to a 100m resolution as inputs to the
NaChem model. The time resolutions of the lidar measurements typically vary between 1 and 10
minutes, depending on the experiment, and are linear interpolated to 0.1 seconds. The reference
profiles inherently include diffusion and other dynamic effects on the sodium species in the MLT, as
these observational data represent snapshots of sodium diffusion at various times. By constantly
matching the observed Na profile to the simulated Na profile, the diffusion is included implicitly in the
model. The seasonal column densities of both ALO and CSU profiles are similar to a sinusoidal function,
with ALO data peaking near June and CSU data peaking in November. The centroid height of the sodium
layer is higher in the ALO data than in the CSU data.

## 4. Results

### 4.1 Sensitivity test

Sodium in the atmosphere could manifest in many forms, i.e., in sodium-bearing neutral chemicals and ionic chemicals. The sodium number densities are typically obtained via lidar measurements. Given the complexity of the sodium chemistry, the observed sodium is merely a subset, possibly not even a major constituent, of the total number of all the sodium-bearing species in the atmosphere. The total sodium content is defined as the total number of sodium atoms in all 14 sodium-bearing species, as listed in Table 2. In summary, the sodium that we can detect does not necessarily provide an accurate representation of the total sodium content or the overall count of sodium-bearing species, as unobservable species such as $Na^+$ and $NaHCO_3$ could constitute a substantial portion of the total sodium content.

Understanding the impact of each background species, i.e., species listed in Figure 3., on the total sodium content is essential to study the underlying mechanism of the chemical reactions. Therefore, we present a sensitivity test by isolating variables. The sensitivity test is done by altering the number density of background species in question by two orders of magnitude, i.e., with a factor of 0.1 and 10, while keeping the number densities of other background species and the atomic sodium fixed. The simulation is kept running until all the numbers are stable. The diurnal variations of the sodium and background species are not considered in sensitivity test as they introduce unnecessary complexity. The results of the sensitivity test of the 11 background species and temperatures involved in the numerical simulation are shown in Figure 3. Each panel contains three lines, where the red curve shows the unaltered vertical profile of the total sodium content. The results of the species altered by the factor of 0.1 and 10 are shown in light blue and yellow, respectively.

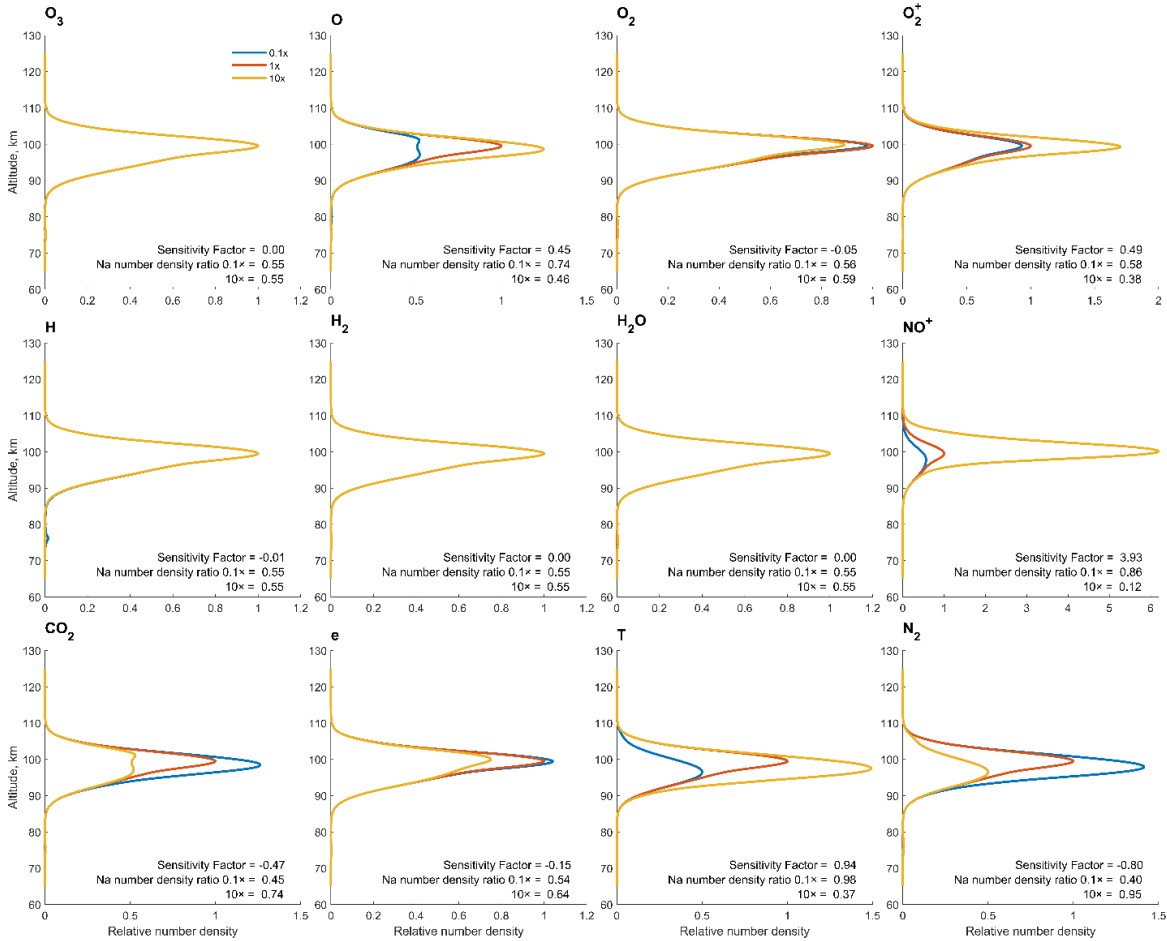

**Figure 3.** *Sensitivity test of 11 background species and temperature on Na chemistry. The total sodium content vertical profile for the respective background species altered by 10x and 0.1x are shown in yellow and light blue. The reference sodium content vertical profiles are shown in red. Additionally, the sensitivity factor and the Na number density ratio to the concentration of all sodium species are presented on each panel.*

In Figure 3, only the yellow curve is visible in some of the panels because the three curves are drawn on top of each other, indicating that the change of the respective background species bears little to no effect on the sodium chemistry. A sensitivity factor is defined to better quantify the weight of each background species in sodium chemistry. The factor is calculated by the following equation:

$$Sensitivity\ Factor = \frac{NaT_c^{10} - NaT_c^{0.1}}{NaT_c} \tag{2}$$

Where $NaT_c^{10}$ is the column density of the total sodium content with the respective species altered by a factor of 10, and $NaT_c^{0.1}$ is the same operation as the previous one but altered by a factor of 0.1. The denominator, $NaT_c$ , is the column density of the reference profile. The sensitivity factor provides a general insight into how variations in the background species correlate with sodium number density. A greater absolute value for the sensitivity factor indicates a stronger correlation. A positive sensitivity factor indicates a positive correlation between the total sodium content and the respective species, and

vice versa. The reference profile is the total sodium content in steady-state in the background condition
of the midnight new year of 2002, giving a typical sodium vertical profile similar to the one shown in
Figure 5 of Plane (2004). In the simulation, a greater total sodium content implies that a smaller
percentage of the sodium chemicals are present as sodium atoms as the altitude profile of the sodium
atoms is fixed in the sensitivity test. In reality, instead of the sodium atoms, the total sodium content
should be more or less conserved. Hence a higher total sodium content in our simulation suggests less
sodium can be detected by the lidar.
Although the sensitivity factor could be different upon the change of the reference profile, it still gives
an insight into the significance of each background species to the sodium chemistry. Apparently, the
weight of some background species, namely $O_3$, $H$, $H_2$, and $H_2O$, is negligible in sodium chemistry,
meaning that removing these species and their associated reactions has no effect on the overall sodium
chemistry. Nevertheless, these species are still kept in our numerical model for completeness. The
impact of species that convert Na atom to $Na^+$, as listed in reactions 27 and 28 of Table 1, is generally
strong. The effect of $NO^+$, in particular, is the most significant according to the sensitivity factor, greater
than the combined effect of all the other species. Consequently, the number density of the observable
[Na] atom by lidar is strongly anti-correlated with the fluctuations of the $NO^+$. In a nutshell, more $NO^+$
will directly lead to fewer observable Na atoms. That being said, the interaction between sodium and
background species is rather complex. The scope of the sensitivity factor in the present paper was
limited to column density. As a result of such, variations and behaviors of the sodium chemicals by
altitude are overlooked. The actual impact of the background species may differ at different altitudes.
**4.2 Meteoric input function**
The estimation of meteoric influx is subject to many uncertainties among different techniques (Li et al.,
2022). Moreover, the meteor flux estimated by the sodium chemistry model also varies (Marsh et al.,
2014; Plane et al., 2015). The previous model of Plane (2004) and the following similar models indicate
that the rate of dimerization, or the speed of removing sodium from the system, is heavily correlated to
the vertical transport in the MLT. The NaChem model does not explicitly incorporate vertical transport,
but the vertical transport by diffusion is inherently embedded within the input of the observed sodium
vertical profile.
Unlike the previous models (Plane 2004; March et al. 2014; and references therein), the present
NaChem model took an indirect route to estimate the meteor mass input. During the simulation, the
$NaHCO_3$ dimerization and the uptake of the sodium species on meteoric smoke particles, which can be
turned on or off, create a deficit of sodium atoms. Meanwhile, a meteor input function injects an
appropriate amount of sodium atoms so that the present sodium vertical profile always matches the
reference profiles. This is carried out by finding the difference between the current sodium profile (with
the deficit) and the corresponding reference profile in every iteration and then replacing the former
with the latter. The study by Plane (2004) found that the diffusion coefficient is highly correlated with
the sodium sink, primarily because the dimerization reaction occurs predominantly at lower altitudes.
The simulation circumvents this uncertainty by directly incorporating the observational sodium vertical
profile, given that diffusion is already inherently in the measurements.

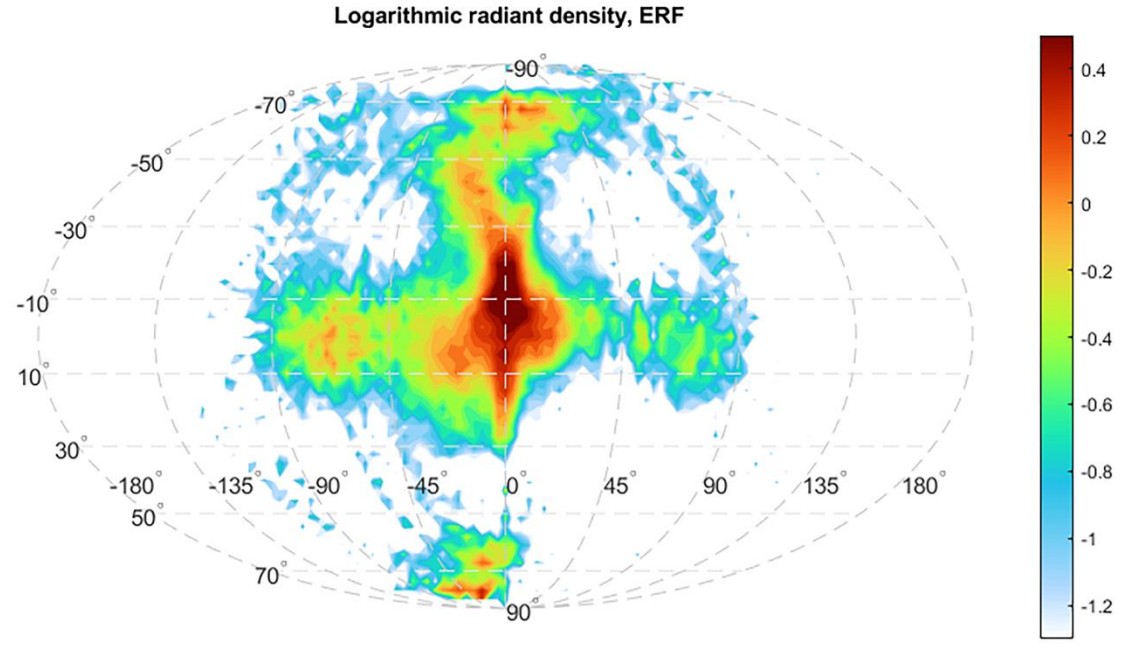


**Figure 4.** *Logarithmic meteor radiant source distribution derived from the AO observations. The figure is in a.u. (arbitrary units). The figure illustrates the relative frequency of meteor occurrence at different radiant directions in the Earth Reference Frame (ERF), equivalent to ground-based observations. The latitude of the ERF is centered on the ecliptic plane. The longitude of the ERF is centered to the Apex direction, the moving direction of the Earth, where the highest number of meteors encounter Earth. The radiant distribution is derived from the number of meteor events. Figure reproduced from Li et al. (2022).*

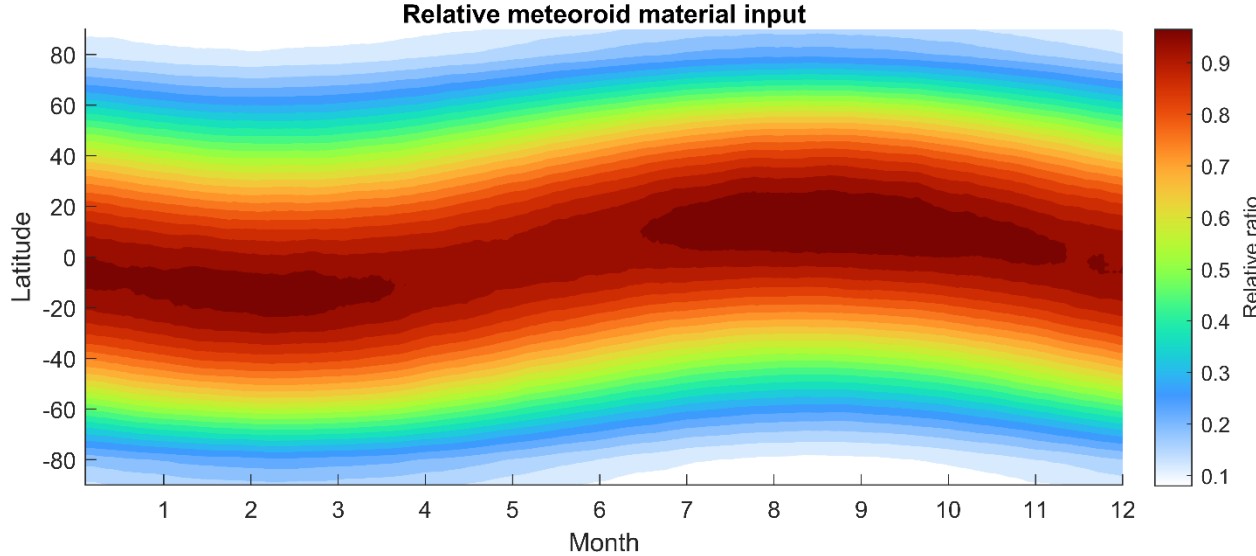

316

**Figure 5.** *Relative seasonal and latitudinal meteoroid input by meteor occurrence, inferred from the radiant source distribution shown in Figure 4. The figure is normalized to its max value.*

319

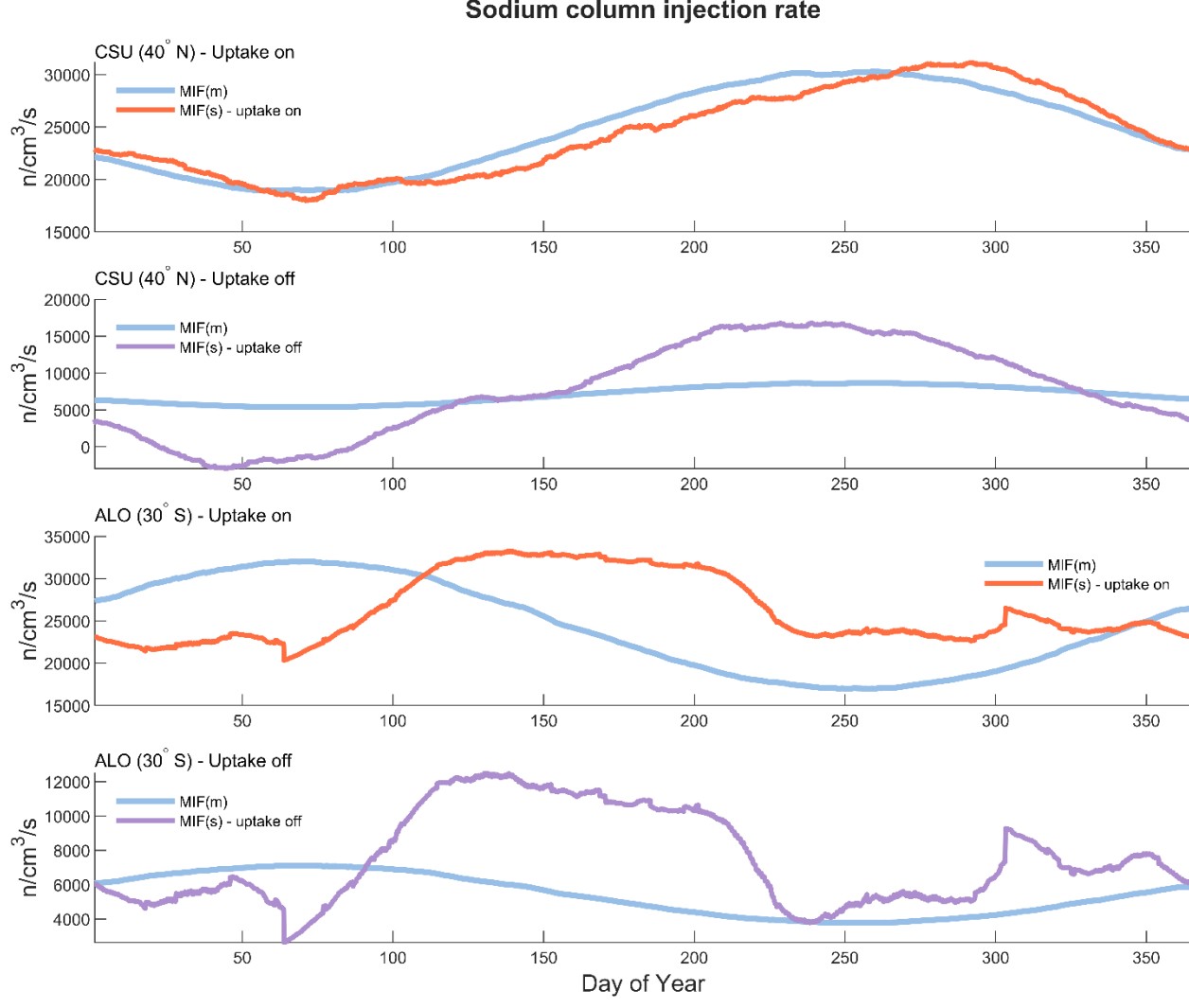

**Figure 6.** *A comparison between two meteor input functions: MIF(m), which is inferred from micro-meteor radiant distribution, and MIF(s), derived from a Na chemistry model with sodium input from lidar observations.*

Figure 4 shows the high-resolution meteor radiant source distribution recently inferred from the AO observations (Li et al., 2022). The typical mass of the Arecibo meteors is estimated to be around $10^{-13}$ kg based on flux rate (Li and Zhou, 2019). Mathews et al. (2001) estimated the limiting meteor mass of $10^{-14}$ kg based on the meteor ballistic parameter. Limiting mass is the smallest mass a meteoroid must have to generate sufficient ionization to be detected by radar. Despite these estimations being based on various simplified assumptions that may lead to inaccurate results, the estimated limiting mass at AO is still at least two orders of magnitude smaller than the estimations of other facilities by similar means. More than 95% of the meteoroid population in the Earth's atmosphere is found to be sporadic meteors by HPLA radar observation (Chau and Galindo, 2007), which typically are low-mass meteors evolved from the outer Solar system due to the Poynting-Robertson drag (Nesvorný et al., 2011; Koschny et al., 2019). That being said, the percentage of sporadic meteors, as well as the radiant source distribution,

are both estimated based on the occurrence. However, the occurrence of sporadic meteors may not be
able to represent their mass distribution. The relative seasonal and latitudinal meteoroid input by the
number of occurrence inferred from the new radiant source distribution is depicted in Figure 5. The
meteoric input generally follows a sinusoidal pattern and differs from the one used in the previous work,
as shown in Figure 1 of Marsh et al. (2013). Although the interplanetary dust (meteor) background on
the Earth's orbit could vary in different locations due to a variety of reasons, e.g., Jupiter resonance, it is
still safe to assume no change in the interplanetary dust background for our purpose. Taking a stable
interplanetary dust background, the MIF(m) 's seasonal sinusoidal pattern should follow the Earth's axis
rotation relative to the ecliptic plane.
Figure 6 shows a comparison between two types of meteoric input function: MIF(m), which is inferred
from the micro-meteor radiant distribution, and MIF(s), derived using the Na chemistry model with
sodium input from the lidar observations. MIF(m) is in arbitrary units and has been linearly scaled to
match the amplitude of MIF(s). Same as MIF(s), MIF(m) is also smoothed by a 15-day running average.
For the MIF(s) model simulations, we did two scenarios, one with and one without uptake by smoke
particles, for the ALO and CSU data. The MIF(s) with uptake by smoke particles exhibit a good match
with the MIF(m) on the CSU dataset, while it does not show as good of a match on the ALO dataset. The
MIF(s) with smoke uptake off is represented by a purple line, while the MIF(s) with smoke uptake turned
on is depicted by an orange line. The MIF(s) could go negative when the reference sodium vertical
profile decreases faster than the removal rate by the dimerization, as shown in the purple line in Figure
6, indicating that the dimerization process alone is not sufficient enough to account for all the sodium
atom depletion in the MLT region. MIF(m) is derived from a global micro-meteor radiant distribution
model, as depicted in Figure 4 and Figure 5. The smoke uptake of sodium species in this study is
implemented using a methodology similar to Plane (2004), but instead of applying smoke uptake solely
to the three major sodium species, namely Na, NaHCO3, and Na+, it is applied to all 14 sodium-bearing
species. The optimal uptake factor to obtain the best results was found to be $2 \times 10^{-2}$/km/s. The smoke
uptake and $NaHCO_3$ dimerization account for approximately 75% and 25% of the Na sink, respectively.
According to the global meteoroid orbital model outlined in Li et al., (2022), the latitudes spanning 29.5°
S to 30.5° S (ALO) account for 0.52% of the total meteor input, while those between 39.5° N and 40.5° N
(CSU) represent 0.67%. The CSU site shares more meteor input due to its closer proximity to one of the
Apex meteor radiant sources. The global total sodium injection rate inferred from the ALO data-based
simulation is $(2.01 \pm 0.68) \times 10^{23}$ atoms per second, and the CSU-data-based simulation suggests a global
sodium injection rate of $(1.28 \pm 0.55) \times 10^{23}$ atoms per second. The error is determined by calculating the
standard deviation of the detrended, unsmoothed raw MIF(s). Note that both MIF(m) and MIF(s)
presented in Fig.6 are smoothed by a 15-day running average. Assuming the relative sodium elemental
abundance in meteoroid material is 0.8% (Vondrak et al., 2008), the deduced total meteoroid material
input of ALO-based simulation is 83±28 t $d^{-1}$. From CSU-based simulation, the rate is 53±23 t $d^{-1}$. Both
estimations are close to 80-130 t $d^{-1}$, the value reported by the Long Duration Exposure Facility (Love
and Brownlee, 1993; McBride et al., 1999). It is worth noting that the estimated total daily input of
meteoroid materials varies among previous studies, ranging from 4.6 t $d^{-1}$ (Marsh et al., 2013) to 300 t $d^{-1}$
$^{-1}$ (Nesvorný et al., 2009), with an intermediate value of 20 t $d^{-1}$ reported by Carrillo-Sánchez et al. (2020).
While these estimates seem quite disparate, the variance is relatively small given that the daily input
rate is derived from combinations of chemicals that can fluctuate by several orders of magnitude. For
example, the $NO^+$, which exhibits the highest sensitivity factor according to the sensitivity test,
undergoes diurnal variations of approximately three orders of magnitude.

**5. Discussion**

The sodium concentration in the sodium layer in the MLT region is governed by several factors, including
chemistry, dynamics, and the MIF. It's difficult to discern which of these three components is more
important than the others. In this section, we discuss various factors that may contribute to modeling
the sodium concentration in the MLT.
The mass of the meteoroids has been estimated and measured using various methods. These include
the ballistic parameter derived from meteor deceleration (Mathews et al., 2001), estimation of meteor
head echo plasma distribution through a combination of meteor ablation models and radar cross-
section measurements (Close et al., 2005; Sugar et al., 2021), flux rate determination (Zhou and Kelley,
1997), as well as spacecraft in-situ measurements (Leinert and Grun, 1990), among others. The mass
estimated by the meteor ballistic parameter is commonly referred to as momentum or dynamical mass.
The mass estimated by the meteor ablation model is usually called the scattering mass. The meteor
momentum mass from Arecibo Ultra-High-Frequency (UHF) radar observation is estimated to be $10^{-14}$ –
$10^{-7}$ kg, with the typical mass being $10^{-13}$ kg. On the other hand, the meteor scattering mass is estimated
to be $10^{-9}$ – $10^{-5.5}$ kg by data from EISCAT UHF radar (Kero et al., 2008) and $10^{-7}$ – $10^{-4.5}$ kg by data from
ALTAIR UHF radar (Close et al., 2005). While the detection sensitivity among different facilities differs,
these estimations are still off by many orders of magnitude. The assessments of either momentum mass
or scattering mass are based on a variety of simplified assumptions. They are subject to errors due to
the complexity of radar beam patterns, background atmosphere conditions, aspect sensitivity, meteor
radiant sources, and many other possible factors. For example, radar meteor observation is subject to
bias against low-mass, low-velocity meteors (Close et al., 2007; Janches et al., 2015).
Another aspect that may contribute to the MIF(m)'s uncertainty is the meteor radiant distribution. The
meteor radiant distributions shown in Figure 4 and many others (Chau et al., 2004; Campbell-Brown and
Jones, 2006; Kero et al., 2012) are inferred or measured by meteor occurrence instead of mass input.
Currently, retrieving a more accurate estimation of the meteor mass input is still a topic under active
research, and there is no quantitative study on the disparities between meteor occurrence and meteor
mass input. The radiant sources of the meteors are expected to differ by mass as their orbital evolution
is highly correlated to their mass. The interplanetary dust interacts with the solar wind while in the Solar
System, losing its momentum in the process and evolving into orbits with a smaller semi-major axis and
lower eccentricity. The effect is called the Poynting-Robertson effect (Robertson and Russell, 1937),
which behaves like a drag force and defines the evolution of interplanetary dust, and it could be the
major reason for the existence of sporadic meteors (Li and Zhou, 2019; Koschny et al., 2019). The
importance of the Poynting-Robertson effect is highly dependent on the density and mass of the object.
By and large, the orbits of the smaller particles evolve exponentially faster. The orbital dynamics of
interplanetary particles have been very well summarized in section 2.2 of (Koschny et al., 2019). For the
reasons above, the meteor radiant distribution of mass could deviate from the radiant distribution of
occurrence. Therefore, the meteor input rates as shown in the blue curves of Figure 6 could be different
from those derived from the meteor radiant distribution of mass since they were derived from the
meteor radiant distribution by occurrence.
In the sodium chemistry model presented in this work, the MIF is the sole source of sodium, while the
sodium sink comprises $NaHCO_3$ dimerization and smoke uptake. The MIF(s) is determined by matching
the sink rate of the sodium atoms with the rate of sodium injection. In other words, MIF(s) represents
the amount of sodium injection needed to keep the sodium concentration equal to the reference
sodium profiles. If the chemical lifetime of sodium in the MLT is short, then the seasonal variation of
both the MIF and sodium concentration in the MLT should be similar. After examining Figures 2, 5, and
6, it can be observed that the averaged seasonal variation of sodium over the years at both sites (ALO
and CSU) does not correspond to the trend of the MIF(m) at their respective latitudes. This may indicate
that the chemical lifetime of sodium in the MLT should be relatively long, as there is no immediate effect
of MIF(m) on the sodium concentration. The MIF(m) displays a sinusoidal pattern which peaks in March
at the ALO's latitude and in August at the CSU's latitude, whereas the sodium layer shows dual peaks in
the CSU's lidar observations and one peak in June in the ALO's lidar observations.
In this study, the MIF(s) derived from the NaChem simulation, based on the CSU lidar measurements
with uptake turned on, was able to match the amplitude of MIF(m) obtained from the meteor radiant
distribution. Although the model does not directly incorporate any dynamical processes, the vertical
transport by diffusion is implicitly included. The model forces the sodium layer to be the same as the
data, which are derived from the average of many years' measurements, in which the diffusions are
inherently embedded. The combination of observational data with the numerical chemistry model in
this paper is a relatively straightforward application of data assimilation (Bouttier & Courtier 2002). The
lidar data of both sites (CSU and ALO) indicate that the sodium column density consistently increases by
about 20% from 22:00 to 4:00 LT the next day. This can be attributed to the fact that, during nighttime,
the large deposits of $Na^+$ formed by daytime reactions slowly neutralized to Na. As a result, the sodium
column density consistently increases throughout the night. The same effect can be reproduced in the
NaChem simulation, albeit with a smaller amplitude. The simulation shows the increase to be about 8%.
The value is obtained by maintaining a constant total number of sodium-bearing species through the
deactivation of the sodium sink.
While meridional transport or atmospheric dynamics both contribute to the seasonal variation of the
sodium layer in the MLT, the diurnal sodium profile is the mean of observations of thousands of days, of
which the variation by atmospheric dynamics should be much less prominent. The lack of explicit
dynamics in the model may be one of the sources of inconsistency when compared to the MIF(m)
observations. Further, the WACCM 6, which supplied the background species to the NaChem, is an older
version that does not fully incorporate the dynamics of each ion species. Despite our results showing
good agreement between the MIF(s) and the MIF(m), there might be several plausible factors that could
lead to potential errors. For example, the Na sink by $NaHCO_3$ dimerization varies by the diffusion rate or
the vertical transport of sodium atoms in the chemistry model (Plane, 2004). Likewise, the MIF(m) may
also differ if the meteoroid mass input differs from the radiant source distribution by the occurrence of
meteors, as discussed in the aforementioned paragraph.
**5. Conclusion**
This work introduced a new sodium chemistry model that simulates the time evolution of all sodium-
bearing species using the continuity equation without making any steady-state assumption. The model
employs an exponential integrator and runs in high-time resolution to maintain numerical stability. The
model is simple to maintain in such a configuration and can be scaled up to include additional
capabilities more easily. The model is highly optimized for processing efficiency and benefits from the
use of an exponential integrator. Therefore, within an acceptable total CPU time, the NaChem can afford
a time resolution of up to milliseconds, several orders of magnitude smaller than those used in other Na
models. During our testing, the CPU time to simulated real-time ratio is about 1 to 1000 using a 0.1
second time step.
The model simulation was able to reproduce the seasonal variation of the sodium layer in the MLT by
simulations of chemical reactions. The simulation results at the CSU's latitude capture the general trend
of the seasonal variation at the location. The MIF(s) based on the ALO data exhibited less conformity
with the corresponding MIF(m), which could be attributed to inadequate statistics of the observational
data. Comparably, the CSU dataset is more reliable as the insufficient lidar hours in the ALO dataset may
lead to inaccurate statistics. In the simulation, when forcing the sodium layer to be the observation-
based reference profile, the inferred MIF is estimated to be $83\pm28$ t $d^{-1}$ at ALO and $53\pm23$ t $d^{-1}$ at CSU.
The numerical simulation by NaChem could reproduce the general trend of diurnal and seasonal
variation of the sodium layer compared to the observations by the CSU Lidar. There are some
inconsistencies in MIF(m) and MIF(s) based on data obtained from ALO Lidar. These inconsistencies may
have originated from poor statistics resulting from insufficient observation hours.
In summary, a new sodium chemistry model has been developed in this work to investigate the
relationship between MIF and the sodium layer. We also compared the MIF(m) derived from radar
meteor observation to the MIF(s) derived from the chemistry model and lidar observations. Our results
indicate that the uptake of sodium species onto meteoric smoke particles removes approximately three
times more sodium than the dimerization of NaHCO3. Our future work will focus on incorporating the
plausible factors that may lead to potential errors discussed above into the chemistry model.


Acknowledgment
The study was supported by NSF Grant AGS-1903346. T.-Y. Huang acknowledges that her work is
supported by (while serving at) the National Science Foundation. Any opinions, findings, and conclusions
or recommendations expressed in this material are those of the authors and do not necessarily reflect the
views of the US National Science Foundation. WF was supported by the UK Natural Environment Research
Council (grant no. NE/P001815/1). The lidar data used in this paper were obtained from The Utah State
University (USU) Sodium LIDAR facility and the Andes Lidar Observatory.

Code/Data availability
The CSU lidar data is available through Utah State University data service (Yuan, 2023). The ALO data is
available through the ALO online database (ALO, 2023). The WACCM data used in this work are available
through Penn State Scholarsphere (Li, 2023b).

Author contribution

Conceptualization, Yanlin L., Tai-Yin H. and Julio U.; methodology, Yanlin L.; software, Yanlin L.; validation, Yanlin L., Tai-Yin H., Fabio V., Julio U. and Wuhu F.; formal analysis, Yanlin L., Tai-Yin H. and Julio U.; investigation, Yanlin L., Tai-Yin H., Julio U. and Wuhu F.; resources, Tai-Yin H., Julio U., Fabio V., and Wuhu F.; data curation, Yanlin L.; writing---original draft preparation, Yanlin L.; writing---review and editing, Yanlin L., Tai-Yin H., Julio U., Fabio V., and Wuhu F.; visualization, Yanlin L.; supervision, Julio U. and Tai-Yin H.; project administration, Julio U. and Tai-Yin H.; funding acquisition, Julio U. and Tai-Yin H. All authors have read and agreed to the published version of the manuscript.

Competing interests

The authors declare no competing interests.

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
