# Peer review of "On the relationship between the mesospheric sodium layer and the meteoric input function"

_Annales Geophysicae, 2023_

## Author Response (AR1)

Comments on the paper on the relationship between the mesospheric sodium layer and the meteoric input function.

The paper is a study of a new sodium chemistry model using the continuity equation without making any steady-state assumption. The model includes all sodium-bearing species and runs at a high temporal resolution of millisecond. The study shows the meteor input functions (MIF) from the meteor radiant and derived from the new sodium model. The study could contribute to simulations of sodium concentration in the MLT region.

Some comments that need to be addressed:

We sincerely thank the reviewer for their valuable feedback. Please find our point-to-point response below:

1) The authors compared the meteor input functions (MIF) obtained between two types of meteor input function in Figure 6. They should provide more comparison of sodium species between the model and the Colorado State University (CSU) and the Andes Lidar Observatory (ALO), which can present the performance of the model.

Response:

The model directly utilizes the lidar measurements to estimate the sodium sink. Consequently, the sodium vertical profiles in the model are precisely aligned with the lidar observations. The ingestion of the lidar observations into the model's estimation process eliminates any variability or differences that would typically arise in a comparison scenario.

2) The authors mentioned the variation by atmospheric dynamics in the discussion. The diurnal sodium profile is the mean of observations of thousands of days, thus the variation by atmospheric dynamics should be much less prominent. As I didn't see comparison of sodium species between model and lidar observations, the influence of meridional transport of metallic sodium ions and atoms on the seasonal variability of sodium number density in the model cannot be considered less prominent.

Response:

As mentioned earlier, the model directly utilizes the lidar observations as reference profiles, as shown in Fig. 2, to guide the simulation for estimating the number of sodium atoms removed from the atmosphere. This estimation, in turn, calculates the amount of meteoroid material input required to maintain the presence of the sodium layer. The reference profiles are obtained directly from the lidar measurements, which inherently include diffusion and other dynamic effects on the sodium species in the Mesosphere and Lower Thermosphere (MLT).

3) How long could the model run steadily since the model is not a self-consistent dynamical transport model? What is the global distribution of the sodium species?

Response:

The total number of sodium atoms is conserved in the model. Therefore, it can run indefinitely with the sodium sink turned off. This study utilizes lidar measurements obtained from the Colorado State University Lidar (CSU) and Andes Lidar Observatory (ALO) to estimate the meteor input function at each site. Subsequently, the global meteoroid material input is estimated based on the relative meteoroid material input derived from the micro-meteor radiant source distribution, as illustrated in Fig. 5. Note that studying the global distribution of sodium species is beyond the scope of the current study.

4) How did the authors conclude that the uptake of sodium species onto meteoric smoke particles removes approximately three times more than the dimerization of NaHCO3?

Response:

The MIF(s) is validated by matching the amplitude/relative ratio of MIF(m). The uptake is found to be approximately three times more than the dimerization of NaHCO3 so that the seasonal variation of MIF(s) would align best with the amplitude of MIF(m).

General comments

This study employs lidar observations of Na in the MLT region in combination with a novel Na chemistry model to estimate the meteoric mass influx into the Earth system. The topic is interesting and suitable for Annales Geophysicae. The paper is in my opinion also relevant for the MLT community and should eventually be published. There are, however, numerous little inconsistensies and unclear statements that should be corrected first (see below)

We would like to express our sincere gratitude to the reviewer for conducting a very thorough review of our work. We have carefully considered and incorporated your valuable feedback into the revised manuscript. We believe the revised version has addressed your comments. Please find our point-to-point response below:

Specific comments

Line 17: "the MIF inferred from the meteor radiant"

I think something is missing here, "meteor radiant" appears incomplete.

Response:

The "meteor radiant" has been changed to "meteor radiant distribution."

Line 35, key point 2: Can you provide error estimates for the daily meteoric input estimates? It's not clear, whether it makes sense to provide 2 digits after the decimal point.

Response:

We agree with the reviewer that having 2 digits after the decimal point does not make sense. The corresponding daily meteoroid material input numbers are rounded to their nearest integer in the key points and in the text.

The error of the daily meteoric material input cannot be accurately determined, as the estimation is largely based on the distribution of meteor radiant sources, which currently lacks an error estimation.

Line 37: key point 3: wording not precise. The reader does not fully understand what this sentence means. Please rephrase.

Response:

Key point 3 has been rephrased to read "The frequency of meteor occurrences might not provide a precise reflection of the mass of meteoroid material input."

Line 49: "Julia et al., 2022"

This reference is missing in reference list. I'm not aware of a colleague with the last name „Julia". Do you perhaps mean "Julia Koch", i.e. this paper:

Koch, J., Bourassa, A., Lloyd, N., Roth, C., and von Savigny, C.: Comparison of mesospheric sodium profile retrievals from OSIRIS and SCIAMACHY nightglow measurements, Atmos. Chem. Phys., 22, 3191–3202, doi.org/10.5194/acp-22-3191-2022, 2022.

Response:

We appreciate the reviewer for pointing out the error. The references have been updated accordingly.

Line 58: "neutral chemistry"

Response:

We appreciate the reviewer for pointing out the typo. The typo has been corrected.

Line 63: „the Meteoroid Input Function (MIF) plays a crucial role"

It would be good to explain in simple words what the MIF is. This may not be clear to all readers.

Response:

A brief explanation of the MIF has been added after the sentence, which reads "The MIF is a function designed to comprehend the impact of the temporal and spatial variability of the meteoroid on the atmosphere (Pifko et al., 2013)."

Line 69: "It's" -> "It is"

Response:

Done.

Line 77: " as evidenced by Arecibo Observatory (AO, 18N 66 W) detecting about 20 times more meteors per unit area per unit time than the Jicamarca Radio Observatory"

A factor of 20 is not "several orders of magnitude", so this does not seem to be a good example.

Response:

We acknowledge that the example mentioned in the manuscript is insufficient to substantiate the point. Therefore, the sentence has been revised to offer more illustrative examples. The modified sentence reads:

"The detection sensitivity varies significantly among different facilities. For instance, the Arecibo Observatory (AO) at 18° N, 66° W detects approximately 20 times more meteors per unit area per unit time than the Jicamarca Radio Observatory (JRO) at 12° S, 77° W, and at least 800 times more meteors than the Resolute Bay Incoherent

Scatter North (RISR-N) radar at 75° N, 95° W, despite all being HPLA facilities (Li et al., 2020, 2023a; Hedges et al., 2022). Of note, meteor flux varies with time and latitude, but the variations cannot account for such a large difference."

Line 80: "Consequently, the radiant mass distribution of the meteors"

Please explain, what the "radiant mass distribution" is, this is not fully clear to me and I think the wording is not very precise.

Response:

The concept of radiant mass distribution is introduced as an alternative to radiant distribution. While the radiant distribution counts the number of meteors entering the atmosphere, the radiant mass distribution represents the mass of meteors entering the atmosphere. The 'radiant mass distribution' is indeed unnecessary and may cause confusion. The sentence has been changed to:

"Consequently, the total mass of the meteors entering the Earth's atmosphere is subject to significant uncertainties."

Line 86: "One of the limitations is that the model cannot reproduce the velocity distribution of the meteors in observations"

Does this mean that a velocity distribution is not considered at all in the model, or that it just cannot be reproduced in a realistic way?

Response:

The meteor velocity distribution is a function of both velocity and time. While the velocity distribution was considered in the flux curve model, the implementation is relatively simple, limiting the model's ability to reproduce the diurnal variation of the meteor velocity distribution, i.e., the meteor velocity distribution is not a function of time in the flux curve model. Meteor velocity distribution can be found in Figure 2 of Li and Zhou (2019).

Reference:

Li, Y., & Zhou, Q. (2019). Velocity and orbital characteristics of micrometeors observed by the Arecibo 430 MHz incoherent scatter radar. Monthly Notices of the Royal Astronomical Society, 486(3), 3517-3523.

Line 93: "Additionally, we compare the MIF obtained from the new sodium chemistry model"

The model alone does not provide the MIF, you also need Na measurements right?

Response:

 Yes, that is correct. The sentence has been modified to include Na measurements. It now reads:

"Furthermore, we compared the MIF derived from the new sodium chemistry model and lidar measurements from CSU and ALO, against the results of the high-resolution meteor radiant distribution recently deduced from the observations conducted at the AO."

Line 110: "Throughout the rest of the paper, the MIF estimated from the sodium chemistry numerical model will be referred to as MIF(s)."

The nature of MIF should be explained in more detail. What is its unit? How many independent variables does it have?

Response:

The definition of the MIF has been added after the said sentence, which reads:

"MIF is a function of time and represents the mass of meteoroid material entering the Earth's atmosphere."

Line 142: "The exponential integrator, expressed in Equation 1, provides the solution to the continuity equation, with the exception of reaction 25 listed in Table 1."

The text is not explicit here: is then the Euler integrator used for reaction 25? Can you briefly explain why you didn't use the same integrator for all reactions? Perhaps I'm missing a point here.

Response:

Reaction 25 is carried out using the Euler integrator because its continuity equation is organized differently from the other reactions. As a result, the exponential integrator shown as Equation (1) does not provide a solution for it. The corresponding sentence has been modified to clarify the point. It now reads:

"The exponential integrator, as expressed in Equation (1), provides the solution to the continuity equation. Notably, reaction 25, listed in Table 1, is an exception and was carried out using an explicit Euler integrator in the simulation. The continuity equation of this reaction contains an additional loss term because it represents the only mechanism apart from the uptakes of sodium species that removes Na atoms from the chemistry simulation. "

Line 145: "that either the exponential integrator or explicit Euler integrator produces nearly identical results"

I think this is not the correct usage of "either ... or"

Response:

We believe that the use of 'either ... or' effectively conveys the intended meaning. Nevertheless, it has been modified for clarity. It now reads:

"Our testing indicates that both the exponential integrator and explicit Euler integrator yield nearly identical results. However, for numerical stability, the explicit Euler integrator requires a step size of ~1μs, which is several orders of magnitude smaller than the exponential integrator."

Line 169: "formerly known as Utah State University (USU) Lidar"

I think it is the other way around. The lidar was first operated at Fort Collins, CO and then in Utah, right?

Response:

That is correct. Thanks for spotting it. The corresponding sentence has been changed to:

"the Colorado State University (CSU, 41.4ºN, 111.5ºW) Lidar, also known as Utah State University (USU) Lidar"

Line 171: ".."

Response:

We thank the reviewer for spotting the error. The extra period has been removed.

Line 171: "It contains a total of 27,930 hours"

"It" here wrongly refers to ALO, because of the previous sentence. Please adjust.

Response:

The sentence has been rephrased to:

"The CSU data comprises 27,930 hours of lidar observations between 1990 and 2020, whereas the ALO data consists of 1872 hours between 2014 and 2019.

Line 184: "The general seasonal trend of the sodium vertical profile retrieved from the CSU lidar observations is similar to the estimation by simulation made by Marsh et al. (2013), whereas the results of ALO lidar observations differ from the Marsh et al. results."

Can you show a Figure to back this up? This statement is quite vague, what is similar, what is different? Or does this refer to Fig. 2?

Response:

The statement does indeed refer to Fig. 2. The sentence has been rephrased for clarity, which reads:

"As depicted in Figure 2, the overall seasonal trend of the sodium vertical profile derived from the CSU lidar observations closely aligns with the simulation-based estimate by Marsh et al. (2013). In contrast, the results of ALO lidar observations deviate from the findings reported by Marsh et al. (2013)."

Line 196: "Finally, we further smooth the profiles by fitting them with a skew-normal distribution using the least squares error method"

What kind of distribution is it? How symmetrical is the actual Na profile?

Response:

The skew-normal distribution is a family of distributions including the normal, but with an extra parameter to regulate skewness (Azzalini & Valle, 1996). The result of fitting the skew-normal distribution closely mirrors the original Na profile. This fitting process is employed to remove noise.

The figure below shows an example of atomic sodium vertical profile from the CSU lidar. The actual Na profile normally exhibits a degree of symmetry.

[Figure]

Fig. 2, both panels, y-axis label: "Km" -> "km"

Response:

Done.

Line 201: "then fitted by a normal distribution that mitigates atmospheric dynamics"

So, is it a Gaussian distribution? Above you wrote "skew-normal"?

Response:

We appreciate the reviewer for pointing out the inconsistency. It should be "skew-normal". The aforementioned text has been updated accordingly.

Lien 206: "The reference profiles used in the NaChem sodium chemistry numerical model inherently account for the effects of diffusion of sodium species as these observational data are the snapshots of sodium in diffusion at any given time."

I read this sentence several times and I don't fully understand its meaning. What does "reference profiles" refer to? The Na "profiles" (measured?) or the background species? If Na: why are Na profiles "used" in the model. The model will simulated Na, right? I suggest rephrasing the statement.

Response:

Although the model is capable of simulating Na, it does not simulate Na in the configuration presented in this work. Instead, the model directly uses measured Na number densities as reference profiles to estimate the sodium sink, i.e., the number of sodium atoms removed by $NaHCO_3$ dimerization (reaction 25 in Table 1), and uptake in the numerical simulation.

The reference profiles are Na lidar measurements fitted using a skew-normal distribution, smoothed by 15-day running average, and subjected to a linear 2-D interpolation across time and altitude. The lidar measurements have an altitude resolution of 500m for ALO and from 75m to 140m for CSU. These measurements are interpolated to a 100m resolution as inputs to the NaChem model. The time resolution is 0.1 seconds.

For clarity, the aforementioned paragraph has been rewritten. It now reads:

"Figure 2 displays the processed annual sodium vertical profiles from the lidar measurements, referred to as reference profiles hereafter. These profiles serve as references to guide the numerical simulation of the NaChem model. The reference profiles are Na lidar measurements fitted using a skew-normal distribution, smoothed by a 15-day running average, and processed through linear 2-D interpolation across time and altitude. The lidar measurements have an altitude resolution of 500m for ALO and from 75m to 140m for CSU. These measurements are interpolated to a 100m resolution as inputs to the NaChem model. The time resolution is 0.1 seconds. The reference profiles inherently include diffusion and other dynamic effects on the sodium species in the MLT, as these observational data represent snapshots of sodium diffusion at various times."

Lines 221 – 223: Isn't this essentially the same statement as two sentences before? I think this is redundant.

Response:

Yes, the sentences are a summary of the previous statements. For a better flow, the sentences have been rephrased to read:

"In summary, the sodium that we can detect does not necessarily provide an accurate representation of the total sodium content or the overall count of sodium-bearing species, as unobservable species such as $Na^+$ and $NaHCO_3$ could constitute a substantial portion of the total sodium content."

Line 224: "impact .. to" -> "impact .. on"

Response:

Done.

Line 224: "to the share of Na atom to the total sodium content"

This statement is misleading and I'm not sure, what the profiles in Fig. 3 actually show. Here you write "the share of Na atoms to the total sodium content" and below you write "total sodium content". This should be clearly explained.

It would also be interesting to mention what the ratio of Na atoms to the concentration of all Na species is.

Response:

The statement and the profiles in Fig.3 refer to the total sodium content. The corresponding sentence has been modified to read:

"Understanding the impact of each background species, i.e., species listed in Figure 3., on the total sodium content is essential to study the underlying mechanism of the chemical reactions."

Fig. 3 has been redone to display the ratio of Na atoms to the concentration of all Na species, referred to as Na number density ratio in the figure. The figure caption has also been modified.

Line 229: "The simulation is kept running until all the numbers are stable."

Are diurnal variations considered here? The WACCM background species will have a diurnal variation, right? Did you stop the simulations at a fixed local solar time? For how many days did the simulation run? Please provide more details here.

Response:

The purpose of this test is to analyze the response of sodium species to individual background species, achieved by isolating variables. Diurnal variation has not been taken into account in the sensitivity test as it introduces unnecessary complexity. The initial conditions are provided by WACCM with time set at 3:00 am on May 24, 2014. This specific time was chosen arbitrarily.

The relevant paragraph has been rewritten for clarity. It reads:

"The sensitivity test is done by altering the number density of background species in question by two orders of magnitude, i.e., with a factor of 0.1 and 10, while keeping the number densities of other background species and the atomic sodium fixed. The simulation is kept running until all the numbers are stable. The diurnal variations of the sodium and background species are not considered in the sensitivity test as they introduce unnecessary complexity."

Line 235: "The total sodium content vertical profile f"

So the relative profile of the total concentration of all Na-species is shown? And not the "share" of Na to all species, as mentioned above?

Response:

The total concentration of all Na-bearing species is shown, not the 'share' of Na to all species.

Line 246: "specie" -> "species"

Response:

Done.

Line 263: "species that converts" -> " species that convert"

Response:

Done.

Line 267: "That being said, the works of the background species are in a rather complex pattern"

Grammar? Sentence incomplete?

Response:

The sentence has been rephrased to:

"That being said, the interaction between sodium and background species is rather complex."

Line 268: "The scope of the sensitivity test in the present paper was limited to column density."

Well, you show profile information in Fig. 3.

Response:

We intended to express that the scope of the sensitivity factor as shown in Eq. 2 was limited to column density. The wording has been changed from 'sensitivity test' to 'sensitivity factor'.

Line 269: "As a result of such, variations and behaviors of the sodium chemicals by altitude are overlooked."

What about Fig. 3?

Response:

The intension was to discuss the limitations of sensitivity factor. The previous sentence has been adjusted accordingly.

Line 281: "and the uptake of the sodium species"

This means uptake on meteoric smoke particles, right? I think this should be mentioned to avoid misunderstandings.

Response:

We thank the reviewers for bringing up the missing information. The sentence has been adjusted to:

"During the simulation, the NaHCO3 dimerization and the uptake of the sodium species on meteoric smoke particles, which can be turned on or off, create a deficit of sodium atoms."

Line 283: "matches the reference profiles"

The reference profiles are the observed profiles, right?

Response:

The reference profiles are lidar measurements processed by skew-normal distribution fitting, running average, and interpolation in altitude and time. A description of the reference profile has been added to the end of section 3.2 Data processing and can be found in one of the replies.

Line 288: "infused"

Correct word? I'm not familiar with this word in this context.

Response:

The sentence has been rephrased for clarity. It reads:

"The simulation circumvents this uncertainty by directly incorporating the observational sodium vertical profile, given that diffusion is already inherently in the measurements."

Lines 291 and 303 : "AO observations"

Please explain / spell out what "AO" means.

Response:

AO means Arecibo Observatory. It is spelled out in line 78.

Line 292: "The result is in the Earth Reference Frame (ERF)"

I'm not really familiar with this frame. What does the longitude in the plot correspond to? I would have expected more or less the same values for all longitudes, but this is apparently not the case?

Response:

The latitude of the ERF is centered ($0^o$) on the ecliptic plane. The longitude of the ERF is centered to the Apex direction, the moving direction of the Earth, where the highest number of meteors encounters Earth. While in some publications the ERF's longitude is centered on the Helion direction, we have chosen to center it on the Apex direction to maintain consistency with the publication we are referencing.

Fig. 4 has been reworked with a different colormap for better presentation. A description of the ERF has been added to the caption. The caption now reads:

Meteor radiant source derived from the AO observations. The result is in the Earth Reference Frame (ERF), equivalent to ground-based observations. The latitude of the ERF is centered on the ecliptic plane. The longitude of the ERF is centered to the Apex direction, the moving direction of the Earth, where the highest number of meteors encounter Earth. The radiant distribution is derived from the number of meteor events. Figure reproduced from Li et al. (2022).

Fig. 6: Why can the MIF with uptake off be lower than the one with uptake on? Does the uptake work in both direction, i.e. is there positive and negative uptake?

Response:

The curves depicting 'uptake on' were adjusted downward to align with the 'uptake off' curves. This adjustment is aimed to demonstrate the difference of amplitude between these curves. We agree that the initial presentation of the figure lacks proper captions and is misleading. Fig.6 has been reworked and now displays separate curves of uptake on and off. The caption of Fig. 6 has also been revised accordingly.

Lines 304 and 306: "Kg" -> "kg"

Response:

Done.

Line 305: "the limiting meteor mass"

What is the "limiting meteor mass"? Something like a detection threshold?

Response:

Yes. Limiting mass is the smallest mass a meteoroid must have to generate sufficient ionization to be detected by radar.

Line 311: "Nesvorn'y"

Response:

done

Line 314: "The relative seasonal and latitudinal meteoroid input by the number of events inferred from the new radiant distribution is depicted in Figure 5."

Is this an approach (to get the „new" radiant distribution) not affected by the issue mentioned in the previous sentence? Or does it suffer from the same problem? This is not clear to me.

Response:

Nearly all radar meteor studies encounter this common issue, including the methodology utilized to derive Figure 5.

Line 324: "uptakes" -> "uptake"

Response:

 Done.

Line 337: "in (Li et al., 2022)" -> "in Li et al. (2022)"

Response:

Done.

Line 342: "Assuming the relative sodium elemental abundance in meteoroid material is 0.8%"

The relative mass ratio is relevant here, not the relative sodium elemental abundance, right?

I'm not able to reproduce the mentioned masses, please explain in more detail how you got these numbers.

Response:

We concur that the term 'relative sodium elemental abundance' is potentially misleading. As a result, we have revised the phrase to read 'relative abundance of sodium in chondritic meteorites.'

The specific ratio of 0.8%, as mentioned in Section 2.4 of the work by Vondark et al. (2008), is referenced to the book by Mason (1971). The relevant references have been added to this manuscript.

References:

Vondrak, T., et al. "A chemical model of meteoric ablation." Atmospheric Chemistry and Physics 8.23 (2008): 7015-7031.

Mason, B.: Handbook of Elemental Abundances of the Elements in Meteorites, Gordon and Breach, Newark, USA, 1971.

Line 346 and line 354: "It's" -> "It is"

Response

Done.

Line 347: "Nesvorn`y"

Response:

Done.

Line 349: "given that the daily input rate is derived from combinations of chemicals that can fluctuate by several orders of magnitude"

What species exactly do you mean here? The Na concentration does certainly not vary by orders of magnitude, right? In this context it would again be interesting to know the ratio of elemental Na to the total Na in all species.

Response:

We are referring to ion-species such as NO+, O2+, and electron density, all of which participate in sodium chemistry. The following sentence has been added after Line 349 (now 366) for clarity, which reads:

"For example, the NO+, which exhibits the highest sensitivity factor according to the sensitivity test, undergoes diurnal variations of approximately three orders of magnitude."

The ratio of elemental Na to the total Na in all species are shown in the revised Fig. 3.

Line 357: "The mass of the meteoroids, which constitute the metal layers"

This statement is not really correct, please rephrase.

Response:

The sentence has been rephrased to read "The mass of the meteoroids has been estimated and measured by various methods"

Line 358: "e.g., ballistic parameter (Mathews et al. 2001); plasma by meteor ablation model, radar cross-section (Close et al., 2005; Sugar et al., 2021), flux rate (Zhou and Kelley, 1997), and spacecraft observations (Leinert and Grun, 1990), to name a few"

I suggest rephrasing this part of the sentence to make the methods more understandable. What is „plasma by meteor ablation model", e.g. ? This is unclear to me. Or „flux rate" ? The meaning is not evident. Does „spacecraft observations" refer to in-situ observations, or, e.g. Na remote sensing measurements from a satellite togehter with modelling?

Response:

The spacecraft observations were in-situ measurements carried out by NASA's Long Duration Exposure Facility (LDEF). However, due to the complexities of meteoroid orbits, the in-situ measurements could be highly biased. This bias is also applicable to other approaches.

The sentence has been rephrased to read:

"The mass of the meteoroids has been estimated and measured using various methods. These include the ballistic parameter derived from meteor deceleration (Mathews et al., 2001), estimation of meteor head echo plasma distribution through a combination of meteor ablation models and radar cross-section measurements (Close et al., 2005; Sugar et al., 2021), flux rate determination (Zhou and Kelley, 1997), as well as spacecraft in-situ measurements (Leinert and Grun, 1990), among others."

Lines 363, 364 and 365: the given mass ranges apply to a single meteor, right?

Response:

Right. The given mass ranges are possible mass of a single meteor.

Line 402: "was consistent"

Consistent in what way?

Response:

The sentence has been rewritten for clarity. "In this study, the MIF(s) derived from the NaChem simulation, based on the CSU lidar measurements with uptake turned on, was able to match the amplitude of MIF(m) obtained from the meteor radiant distribution."

Line 404: "diffusion would have been implicitly included"

"would have been" or "is"?

Response:

It should be "is". The sentence has been modified to read: "Although the model does not directly incorporate any dynamical processes, the vertical transport by diffusion is implicitly included."

Line 408: "The lidar data of both sites (CSU and ALO) indicate that the sodium column density consistently increases by about 20% from 22:00 to 4:00 LT the next day"

Good point! At what LST do you determine MIF(s)? How do you deal with the 20% variation?

Response:

The MIF is determined by the average value between 22:00 to 4:00 LT, which corresponds to the time window of the lidar measurements. The 20% variation was averaged into one altitude profile for each day in the reference profiles as shown in Fig. 3.

Line 413: "This number is obtained by turning the sodium sink off and keeping the total number of sodium in the system conserved."

I'm not sure this is an apples-to-apples comparison?

Response:

The sentence has been changed to:

"The value is obtained by maintaining a constant total number of sodium-bearing species by turning off the sodium sink."

Line 434: "During our testing, the CPU time to simulated real-time ratio is about 1 to 100 using a 10-millisecond time step."

But this high time resolution was not used here, right? Above you wrote that the default time step is 0.1 seconds.

Response:

That is correct.  The high-time resolution was not used here. The sentence has been changed to "is about 1 to 1000 using a 0.1 second time step" for consistency.

Line 436: "The model simulation was able to reproduce the seasonal variation of the sodium layer in the MLT by simulations of chemical reactions. The simulation results at the CSU's latitude capture the general trend of the seasonal variation at the location."

Please include a Figure showing that.

Response:

The NaChem model was able to capture the general trend of the seasonal and diurnal variation using simulations of chemical reactions as shown in the figures below. However, we do not include these figures in the manuscript for the following reasons:

    a.  Simulations of seasonal and diurnal variations were conducted for the purpose of model validation. The results are depicted in the figures below. The initial conditions were given by lidar measurements, and the background conditions were obtained from WACCM data. These simulations do not incorporate vertical or meridional transport. Furthermore, due to the absence of active meteoroid material input in the model, the simulation was run with the sodium sink turned off. As such, with no sodium production and sink, this configuration and its results are not realistic.

b.  The simulation that produced the results presented in Fig. 6 was guided by Na reference profiles obtained from lidar measurements. Displaying results, e.g., the figures below, that were not guided by such reference profiles could be misleading.

[Figure]

Line 442: "The numerical simulation by NaChem could reproduce the general trend of diurnal and seasonal variation of the sodium layer compared to the observations by the CSU Lidar."

This should also be backed up by showing a figure.

Response:

Please refer to the figure below. We chose not to include these figures for the same reasons mentioned in the previous comment.

[Figure]

Line 467: "All authors have equal contributions to the work."

I suggest being more specific here. Probably not all authors contributed equally to the data processing, analysis, writing etc.

It has been updated in the manuscript.

Reference list: References not in accordance with Ann. Geo. standards, please update. The list also contains several typos. Please check carefully.

We have updated the reference list.

---

## Author Response (AR2)

Reviewer #2

Response to the reviewer

General comments:

I thank the authors for considering the comments in my previous review. I apologize, but I only realized after reading the revised version of the manuscript that there may be a fundamental problem with your basic approach. My interpretation may be wrong and if that is the case, please let me know.

In the initial version of the manuscript the MIF was not really defined (and it is still not described well in the revised version). My earlier understanding was that MIF corresponds to a model of the daily meteoric mass influx as a function of latitude (and perhaps longitude) with units of, e.g. kg / day. But this is not the case, I think? If I understand your approach correctly, you are matching (in Fig. 6) the seasonal variations of MIF(m) and MIF(s), but not the absolute values of the MIFs. These two MIFs in fact have different units. MIF(m) is dimensionless (?) and scaled to MIF(s) in Fig. 6, right? MIF(s) on the other hand has the units /cm^3/s. My initial understanding was that both MIFs have the same units and are matched here, but this apparently is not the case. Then the question is, what criterion is actually used to do the matching. I think that you are trying to reproduce the seasonal variation of the MIFs – is this correct? I'm not sure this is a really robust approach.

Another problem is related to the estimated of the total daily meteoric mass influx. As I understand it, these values cannot really be robustly estimated using your approach, because you empirically adjust the uptake factor (for the uptake on meteoric smoke particles) to obtain the "best results". It is not explained at all what "best results" means. For the reader this means, that the uptake factor is adjusted in some arbitrary way. The problem now is that your estimate of the meteoric mass influx depends directly on the chosen uptake factor. In other words, your estimated mass influx is not really robust, right?

If my interpretations are correct, then there are some fundamental flaws in the approach taken in the manuscript and it would then be very difficult to interpret the meaning of the results. If I'm wrong, please let me know.

Response:

We thank you for the opportunity to further clarify the points that were raised by you. No need to apologize. There is some misunderstanding (no fault on you). Below please find our responses to your questions. We believe they adequately address your questions or concerns. Thank you again for your input to make this article better.

The MIF(m) is a relative unitless quantity, representing the relative seasonal changes of the meteoroid material input from radar measurements.  Therefore, MIF(m) provides the insights of the relative seasonal variation of the meteoroid material input. The mass of the meteoroids cannot be accurately determined from radar measurements. The uncertainties in estimating meteor mass from radar measurements are discussed between line 386 and line 402 in the revised manuscript.

On the other hand, MIF(s) derived from the chemistry model has a unit of $1/cm^3$/second of sodium atom, which can be converted to meteoroid mass input in $Kg/m^3/s$ with estimations in relative sodium elemental abundance of 0.8% (line 371) in meteoroid material.

The goal of this study is to match the amplitude of seasonal variation of the MIF(s) and MIF(m). In other words, the value of [MIF(s)/mean(MIF(s))] should be similar to MIF(m). In essence, we are comparing MIF(m) and MIF(s) but given that the former one is a relative quantity and the latter one's value can be computed, we need to scale MIF(m) by multiplying the mean value of MIF(s) so that these two quantities can be compared, and the least square error can be found. The value of MIF(s) can be determined from the numerical chemistry simulation. Fig.6 is presented as [MIF(m) x mean(MIF(s))] and MIF(s), in this form it conveys more information.

As illustrated in Panel 2 (CSU 40° N – Uptake off) and Panel 4 (ALO 30° S – Uptake off) of Figure 6, the amplitudes of MIF(s) and MIF(m) do not align without uptake. Notably, MIF(s) becomes negative around day 50 due to the faster depletion of sodium compared to the sodium sink rate caused by NaHCO3 dimerization. This is physically impossible, necessitating the consideration of uptake. We believe this approach is innovative and provides a rather robust way to infer MIF.

The amplitude of the MIF(s) can be adjusted by altering the uptake rate. Please refer to the figures attached at the end of this document. They show the matching with different uptake factors, and the results should be self-explanatory.

Specific comments:

Line 64: "to comprehend" -> "to describe" or "to model" ?

Same sentence: Thanks for adding this information. I suggest adding information on the units of the MIF and the variables it depends on (Latitude, time, longitude? .. etc.?)

Response:

"to comprehend" has been replaced by "to model".

More description on MIF and the variables has been added to line 117-122, where MIF(s) and MIF(m) are first introduced. The corresponding text reads:

"The MIF(m) is determined through a 3-D meteoroid orbital simulation, a process similar to the seeding process discussed in section 3.1 of Li et al. (2022), based on the meteor radiant distribution. MIF(m) is a relative unitless quantity. Note that the meteor mass cannot be accurately determined via radar measurements, however, the seasonal variation of meteoroid material input can be represented by MIF(m). The estimation of meteor mass is further discussed in Section 5. In contrast, MIF(s) is expressed in units of $1/cm^3$/second."

Line 81: "Of note" -> "Of note is that" ?

The corresponding sentence has been rephrased to:

"While meteor flux does exhibit variations based on time and latitude, these fluctuations alone cannot explain the magnitude of the observed difference."

Line 111: "This study estimates the MIF in the numerical model by matching the dimerization reaction to maintain the observed sodium present in the MLT."

This is one of the general points that I do not understand. Do you estimate the absolute value of the MIF as a measure of the daily meteoric mass input into the Earth system? See also the general comment above and the comments below. I think that the paper does not fully and explicitly describe what was done.

Response:

This study obtains the value of MIF(s) from the chemistry model (NaChem) and compares the relative value of MIF(s) to MIF(m). The process has been discussed to a greater extent in the response to the general comments. The sentence mentioned in this comment has been revised for clarity:

"This study estimates the MIF in the numerical model by matching the amount of sodium atoms removed by the dimerization reaction and uptake, i.e., sodium sink, to maintain the observed sodium presence in the MLT."

Line 116: "The MIF(m) is determined through a 3-D meteoroid orbital simulation based on the meteor radiant distribution."

Is there a reference with the details of this procedure? And: what are the units of the MIF(m).

Response:

MIF(m) is a unitless quantity. The sentences have been rewritten and have been included in the previous answer.

Line 120: "The background major gas species, including O3, O2, O, H, H2, H20, etc., and the temperature are provided by WACCM"

WACCM O3 is/was known to be systematically too low in the MLT (by about 50%). Is this also the case for the WACCM version used here? Which version is used? If there is a systematic bias in WACCM O3 (and potentially other species), how would this affect your simulation results?

Response:

The model used was CESM2 with WACCM6. Regardless which version is used, $O_3$ has virtually no effect on sodium chemistry as shown in Figure 3. The WACCM version has been added to the corresponding text.

Equation (1): I think the units don't fit here. c and x_0 are concentrations? Or is (1) a numerical value equation (where the units are ignored)?

Response:

Eq. 1 is a first-order numerical exponential integrator for numerical simulation. It is a solution of the continuity differential equation, which does not have a unit. The exponential integrator is utilized to mitigate the problem of numerical instability.

Line 144: "Where $x0$ is the value"

Is it the concentration (with units 1/cm^3)?

Response:

Yes. In simulation, $x_0$ is the number density of the species.

The corresponding sentence has been expanded to: "Where $x_0$ is the value in the current step. In the simulation, it is the number density of the species."

As mentioned in the previous response, Eq.1 represents a numerical integrator, which does not have a unit.

Line 149: "apart from the uptakes of sodium species"

"uptakes" -> "uptake" ? You mean the uptake on meteoric smoke particles?

Response:

There are 14 sodium species (as shown in Table 2) in the simulation. The uptake of each species is calculated independently by an additional term in their continuity equation. E.g., Eq. (2), Eq. (3) and Eq. (4) in Plane (2004).

Plane, J. M. C. "A time-resolved model of the mesospheric Na layer: constraints on the meteor input function." Atmospheric Chemistry and Physics 4.3 (2004): 627-638.

Line 166: "The [hv] is the term that represents loss via photon emission"

?? Emission? It is absorption followed by dissociation for most of the reactions listed in the table, not emission, right?

Response:

You are right. It should be absorption. The sentence has been revised to:

"The [hv] is the term that represents the loss via photoionization, which is approximately a sinusoidal function based on the Solar zenith angle of the respective local time.

Same sentence: "which follows a sinusoidal function"

Is it really a sin function? sin(SZA = 0) = 0, i.e. for the sun in the zenith the term would vanish. This is not the correct dependence, right?

Same sentence: "zenith angle" -> "solar zenith angle"

Response:

It is approximately a sinusoidal function because it involves the Earth's rotation.  The definition of a sinusoidal function includes both the sine and cosine function.

Same sentence: "local time" -> "local solar time"; In addition, the SZA does not only depend on local solar time, but also on latitude.

Response:

"local time" has been changed to "local solar time". The latitude information is represented by the solar zenith angle.

Line 191: "In contrast, the results of ALO lidar observations diverge from the findings reported by Marsh et al. (2013)."

Diverge in what way?

Response:

The corresponding texts have been expanded to:

"In contrast, ALO lidar observations deviate from the findings reported by Marsh et al. (2013). The ALO measurements exhibit a prominent peak around June, while the results in Marsh et al. (2013) show a double peak in March and October."

Line 216: "The time resolution is interpolated to 0.1 seconds."

What does this mean? Is the actual time step different? If yes, what is it?

Response:

The text has been revised to:

"The time resolutions of the lidar measurements typically vary between 1 and 10 minutes, depending on the experiment. In this study, they are linearly interpolated to 0.1 seconds."

Line 249: "Additionally, the sensitivity factor and the Na number density ratio to the concentration of all sodium species .."

Is the density ratio averaged over all altitudes or given for a specific altitude? This should be mentioned.

Response:

The curves presented in Fig.3 contain altitude information. The sensitivity factor, on the other hand, is calculated with column density. This has been discussed in the text below Eq.2.

Line 256: "The factor is measured": "measured" -> "determined" or "calculated" ?

Response:

The word has been replaced by "calculated'.

Line 260: "For example, a Sensitivity Factor of 5 indicates that the total sodium content increases by five times when the respective background species increases 100 times."

This is not correct, because the sensitivity factor is not normalized by the value for a factor of 0.1, but for the reference value.

Assume, e.g., the following: NaTc10 = 5, NaTc0.1 = 1, NaTc = 2. Then there is a factor of 5 increase, but:

Sens. factor = (5 - 1)/2 = 2 !

Response:

You are correct! Thanks for spotting it. We do want to emphasize that this does not affect the outcome of the sensitivity test. The corresponding text has been changed to:

"The sensitivity factor provides a general insight into how variations in the background species correlate with sodium number density. A larger absolute value for the sensitivity factor indicates a stronger correlation. A positive sensitivity factor indicates a positive correlation between the total sodium content and the respective species, and vice versa."

Line 267: "as the altitude profile of the sodium atoms is fixed."

? Why is it fixed? I thought the model is used to calculate the [Na] profiles, among profiles of other species?

Response:

The altitude profile of [Na] is fixed only in the sensitivity test. In essence, this test calculates the ratio of [Na] in all sodium-bearing species (e.g., those listed in Table 2) at the equilibrium state for a given background condition. While the altitude profiles of other sodium species vary with the background condition, [Na] remains fixed, serving as an unlimited supply and sink of sodium atom.

The corresponding text has been modified to clarify:

"In the simulation, a greater total sodium content implies that a smaller percentage of the sodium chemicals are present as sodium atoms because the altitude profile of the sodium atoms is fixed in the sensitivity test."

Line 278: "is strongly correlated"; or rather "is strongly anti-correlated" ?

Response:

It should be anti-correlated. The corresponding text has been changed.

Line 298: "The diffusion coefficient is found to be highly correlated with"

Which diffusion coefficient? Not the one in your model, because you don't consider diffusion. Does this refer to WACCM? And shouldn't the diffusion coefficient be independent of some species concentration or removal rate?

Response:

It was referred to the finding in the reference provided at the end of the sentence. The diffusion coefficient is highly correlated with the sink rate because the sodium sink mostly occurs at lower altitudes (<90km). Therefore, the amount of sodium removed by [$NaHCO_3$] dimerization is strongly connected to the amount of [Na] atom transferred to lower altitude via diffusion. The corresponding text has been revised to avoid confusion:

"The diffusion coefficient is found to be highly correlated with the sodium sink due to the dimerization reaction mostly occurs at lower altitudes. (Plane, 2004)."

Fig. 4: What is shown here? What are the units of "radiant density"? The colour bar has not units. Excuse my ignorance, but I don't understand what this plot displays and it is not well explained.

Response:

The radiant density is a unitless relative quantity. The caption has been revised to:

*"Figure 4. Logarithmic meteor radiant source distribution derived from the AO observations. The figure illustrates the relative frequency of meteor occurrence at different radiant directions in the Earth Reference Frame (ERF), equivalent to ground-based observations. The latitude of the ERF is centered on the ecliptic plane. The longitude of the ERF is centered to the Apex direction, the moving direction of the Earth, where the highest number of meteors encountering Earth. The radiant distribution is derived from the number of meteor events. Figure reproduced from Li et al. (2022)."*

Fig. 5: Does "meteoroid input" correspond to the mass or the occurrence?

Response:

Meteoroid input in Fig. 5 corresponds to the occurrence. The caption has been revised for clarity.

"Figure 5. Relative seasonal and latitudinal meteoroid input by meteor occurrence, inferred from the radiant source distribution shown in Figure 4."

Fig. 6: The MIF(m) curves (blue) in the top two panels should be identical, right? This is not the case?

OK, the caption says that the blue curves are linearly scaled, but why? The absolute differences are also important, right?

With this scaling, the reader does not know what the actual MIF(m) value is. The model should be judged by its agreement with the "measurement", right? Same comment for the bottom two panels.

Response:

The MIF(m) in Fig. 6 is scaled with the mean of MIF(s) by multiplying the mean of MIF(s). This procedure allows us to compare the scaled MIF(m) to MIF(s). As was mentioned several times in the previous responses, our study compares the curve of MIF(s) and MIF(m). Please refer to the figures at the end of this response.

Fig. 6, bottom panel: What are the discontinuities in MIF(s) caused by, e.g. around day 65 and around day 305? Are they expected?

Response:

The discontinuities result from missing days in lidar observations, because the ALO data is relatively sparse compared to the CSU data. As shown in Fig. 1, day 65 and day 305 are both at boundaries of the missing lidar days.

Line 323: "the estimated mass"

Which mass do you mean? The limiting meteor mass or the daily or annual meteoric mass input into the atmosphere?

Response:

The limiting mass of the Arecibo Radar. The corresponding sentence has been revised for clarity.

"Despite these estimations being based on various simplified assumptions that may lead to inaccurate results, the estimated limiting mass at AO is still at least two orders of magnitude smaller than the estimations of other facilities by similar means."

It looks like a solid line, not a dotted one.

Response:

It is indeed a solid line. The word 'dotted' has been removed.

Line 350: "The optimal uptake factor to obtain the best results was found to be 2x10^-2/km/s."

How was this determined? What is the best result? This seems like an arbitrary scaling and I'm not sure what the results then really mean? Since the absolute value of the meteoric mass influx directly depends on the uptake factor, your mass estimates are not really robust, right? Perhaps I'm missing a point, but this seems to be a major weakness of your approach?

Response:

The optimal uptake factor is found by the value that leads to the smallest least square error between [MIF(s)/mean(MIF(s))] and MIF(m). It was not an arbitrary scaling as there exists an optimal solution, where the least square error between MIF(s)/mean(MIF(s)) and MIF(m) is minimized. Please refer to the figures at the end of this response for better illustration.

Lines 356 following: The estimates presented here will directly depend on the uptake factor, right? In other words, you can scale the estimated Na influx by altering the uptake factor, which was chosen in a somewhat arbitrary way ("to obtain the best results" and it is not explained at all, what the best results are). Perhaps I am missing a point, but what can you actually learn about the daily Na input from your study? As I understand it now, the estimated are not robust. Please tell me if I'm wrong!

Response:

As was responded earlier, it is the optimal, but not an arbitrary, uptake factor that was found. Our approach provides a robust estimation of Na influx by comparing the relative seasonal variation of MIF derived from the modeling, lidar, and radar measurements. (MIF(s) from lidar measurements and numerical modeling; MIF(m) from radar measurements).

Line 391: "The meteor radiant distributions shown in Figure 4 and many others (Chau et al., 2004; Campbell-Brown and Jones, 2006; Kero et al., 2012) are inferred or measured by meteor occurrence instead of mass input."

So, does this mean, that you don't attempt to reproduce the absolute value of the MIF(M) by the model, but only the seasonal variation? This is not really clearly stated in the paper (the MIF is not well explained and no units are given) and I'm only realizing this now. Perhaps I'm missing a point, but if this is the case, this would be a major weak point of the study.

Please explain in simple words, what was actually done and where the information to quantify MIF(s) actually comes from. If my understanding is correct, your approach does NOT allow estimating the meteoric mass input.

Response:

Since MIF(m) is a unitless relative quantity, its actual value cannot be produced. Measuring the actual value of MIF(m) might be possible, the community is still working towards this goal.

MIF(m) offers insights to the relative seasonal variation of the meteoroid material input based on radar measurements. Our study aims to reproduce this relative seasonal variation by MIF(s), which combines [NaHCO3] dimerization and uptake. Please refer to the figures at the end of this document for better illustration. As shown in Response Fig. 1 (at the end of this document), which illustrates the square difference between [MIF(s)/ mean(MIF(s)) and [MIF(m)], there exists one optimal solution.

Line 410: "The MIF(s) is determined by matching the sink rate of the sodium atoms with the rate of sodium injection."

Do you really have absolute values of the sodium injection? My understanding now is that this is not the case?

Response:

This sentence is to indicate that the absolute value of MIF(s) is derived from the numerical chemistry model based on lidar Na measurements.

Line 422: "was able to match the amplitude of MIF(m)"

Only the relative amplitude, not the absolute amplitude, right? And it is the amplitude of the seasonal variation, if I understand correctly?

Response:

That is correct.

 "Further, the WACCM, which supplied the background species to the NaChem, is an older version that does not fully incorporate the dynamics of each ion species."

Which WACCM version is it? Is there a version number?

Response:

WACCM6 in CESM2 version. The WACCM version is now specified in line 126.

 "Despite our results showing good agreement between the MIF(s) and the MIF(m)"

Not in absolute terms, only regarding seasonal variations, right?

Response:

Correct. MIF(m) cannot be matched in absolute terms.

 "Our results indicate that the uptake of sodium species onto meteoric smoke particles removes approximately three times more sodium than the dimerization of NaHCO3."

I don't think this really is a robust result. If you "empirically adjust" the uptake rate to get the "best results"? As I said, perhaps I am missing a point, but I think this conclusion cannot be drawn.

Response:

Now we understand why you think our approach is not robust because you think that we arbitrarily adjust the uptake to get the best results. This is not quite the full story or the right interpretation of our approach. As we mentioned/responded earlier, the uptake was not adjusted arbitrarily. It was found when the matching of the measured and simulated MIF has the smallest least square error. This ensures that the uptake needed for the best match is indeed the one for the best match. Please refer to the figures below for detailed illustrations.

[Figure]

**Response Fig. 1**. This figure illustrates the square Error between MIF(s)/mean(MIF(s)) to MIF(m). It`s clear that there is an optimal solution at about $2x10^{-2}$. The square error is in arbitrary units.

MIF(m) is a unitless relative quantity, in other words, we are comparing the relative changes of MIF(s) and MIF(m) across different time periods. The MIF(s) is divided by its mean for easier comparison, and this operation does not affect the relative change of MIF(s) across different time periods.

To further illustrate the point, please refer to the Response Fig. 2.

[Figure]

**Response Fig. 2**. This figure depicts the comparing of [MIF(s)/mean(MIF(s))] (in orange) and [MIF(m)] (in blue) with different uptake factors. The y axis is relative strength, and the x axis is day of year. The uptake factor and the square error between MIF(m) and MIF(s) are listed on each panel. Among these figures, it's clear that when the uptake factor is $1.93\times10^{-2}$, the square error reaches its minimum.

After obtaining the best fits between MIF(m) and MIF(s), Fig. 6 in the manuscript is presented with mean of MIF(s) multiplied back to the curves, as the mean of MIF(s) also contains the information of the absolute values of the sodium injection rate. For example, with uptake on, the two curves on the first tile of Fig. 6 are centered at about 24000 n/cm$^3$/s. On the other hand, with uptake off, the two curves on the

second tile of Fig. 6 are centered at about 6000 n/cm$^3$/s. The absolute values of mean(MIF(m) illustrate that the sodium removal due to uptake is approximately three times the effect of dimerization.

---

## Author Response (AR3)

General comments:

The authors have improved the description of the basic approach employed here, particularly the difference between the two MIFs used. However, there is still a lack of information on the exact procedure, especially related to the results presented in Fig. 6 (see also the specific comments below). And I still think that you should provide error estimates for the daily meteoroid mass influx into the Earth system. Without error estimates the relevance of your estimates remains unclear.

Response:

We thank the reviewer for the comments. We further clarified the solving procedures and included the error estimations. Please see the point-to-point responses below.

Specific comments:

Lines 35/36, bullet point 2: Please provide an uncertainty estimate for the values given.

Response:

The uncertainty estimation is now provided.

Line 64: „The MIF is a function designed to model the impact of the temporal and spatial variability ..“

I suggest replacing this by: „The MIF is a function designed to model the temporal and spatial variability ..“, i.e. delete "impact of the"

Response:

Done.

Lines 117 – 122: Thanks, this is now very clear!

Equation (1): I still don't understand, how the units of the different terms in this equation are treated. What are the units of a_0 and b_0? They have the same units, i.e. x_1 is dimensionless. Does this make sense?

Response:

Eq. (1) is a first-order exponential integrator for numerical simulation. The units are consistent with the unit of the rate coefficient used in the numerical model. In our case, the units of $x_0$ and $x_1$ are $1/cm^3$. The units of $a_0$ and $b_0$ are $1/cm^3/second$.

Line 171: „Na(2p)" -> „Na(^2P)"

Response:

Na(2p) has been changed to Na($^2$p) in Table 2 and in the caption.

Line 187: The figure caption is not precise. Not the data is shown, but the available hours of observation.

Response:

The figure caption has been replaced by 'Available hours of lidar observations. CSU lidar (1990-2020, upper plot) and ALO lidar (2014-2019, lower plot).'

Line 209: „Finally, we further smooth the profiles by fitting them with a skew-normal 209 distribution"

This is a vertical smoothing, right? Please mention this explicitly, otherwise it is not entirely clear (to me, „profiles" are always vertical profiles, but the term is also used for time series etc.).

Response:

It is a vertical smoothing. The sentence has been revised to "Finally, the vertical profile for each time step is further smoothed by fitting it with a skew-normal distribution (Azzalini & Valle, 1996), using the least squares error method."

Line 224: „The time resolutions of the lidar measurements typically vary between 1 and 10 minutes, depending on the experiment, and are linear interpolated to 0.1 seconds."

? Is the variation of the lidar data throughout the night used at all? My understanding was that you use nightly averaged data?

Response:

The lidar data variation throughout the night is included. The averaged lidar measurements are then interpreted to a resolution of 0.1 seconds to be consistent with the numerical simulation's time resolution.

Line 229: „The seasonal column densities" -> „The seasonal variation of the column densities .. IS similar .."

Response:

Done!

Line 266, equation (2): This is not very intuitive. It would have been better to define the sensitivity factor as a ratio of the Na columns, not the total Na content columns (and keeping total Na constant, rather than Na). I don't really understand the motivation for this choice.

Response:

The goal of the sensitivity factor is to shed light on the significance of each background species to sodium chemistry. The ratio of Na columns remains the same whether Na or total Na content is held constant in the steady state. We believe that keeping Na constant is more intuitive, as Na can be directly measured, whereas the total Na content cannot be measured at present.

Line 308: „The diffusion coefficient is found to be highly correlated with the sodium sink due to the dimerization reaction mostly occurs at lower altitudes."

This is not "found" in your study, because diffusion is not considered, right? The statement is misleading and the grammar is also not correct. Please improve.

Response:

The sentence has been revised to "The study by Plane (2004) found that the diffusion coefficient is highly correlated with the sodium sink, primarily because the dimerization reaction occurs predominantly at lower altitudes."

Line 313: „Logarithmic meteor radiant source distribution ..."

Ok, it is a logarithmic quantity, but what are the units of the original data? I'm not an expert and this is not clear to me. Do the absolute values have any meaning? I don't want to be picky, I just want to understand what exactly is shown here and this is not well explained.

Response:

The result is in arbitrary units. Therefore, the absolute values carry no meaning. The unit of this figure is [1/unit time/cell of equal area]. It means that the radiant directions in color of $0.5$ ($10^{0.5}$) encounters 30 times of meteors compared to the region in color of -1 ($10^{-1}$) per unit area per unit time. The units have been added in the caption.

Line 320: „Relative seasonal ..."

Relative or normalized to what? To the maximum value? Please mention this explicitly.

Response:

The values shown in the figure were normalized to its maximum value. The caption has been revised to explicitly state this for clarity.

Text discussing Fig. 6: It should be mentioned here again, that MIF(m) is actually dimensionless and is manually (?) scaled here to match the values of MIF(s)? How is this matching done?

Response:

The matching is done exactly like what you described. The goal of the matching is to minimize the difference between MIF(m) and relative MIF(s), i.e., finding the smallest least squared error during the matching process.  This process is solved algorithmically.  The results from the solving process are illustrated in the figures attached to the responses of the previous revisions. Please see the figures in the previous response.  The Response Fig. 1 depicts the difference between MIF(m) and relative MIF(s), and the Response Fig. 2 shows several cases of the uptake value used and the corresponding least squared error. As you can see, this process will find an uptake factor that minimizes the square error between the MIF(m) and relative MIF(s). This is a very standard widely used numerical procedure.

Fig. 6, top two panels: ? I don't understand why the relative seasonal variation of MIF(m) depends on whether the Na uptake on meteoric smoke is switched on in the model or not. MIF(m) - at least its relative variation - should be independent of the model?? It is not clear, what was actually done here.

Response:

The relative seasonal variations of MIF(m) on the top panels of Fig.6 were the same. The panels in Fig. 6 are not on the same scale.

MIF(m) in panel 1 is between 30281 and 18919. MIF(m) in panel 2 is between 8652 and 5406. The maximum to minimum ratio for both MIF(m) in the top two panels is about 1.6.

Fig. 6, bottom two panels: The relative amplitude of the seasonal variations in MIF(m) is also different between the two cases (uptake on/off) for ALO. This should not be the case, right?

Response:

The relative amplitudes of MIF(m) for the lower two plots are the same.

MIF(m) in panel 3 is between 32025 and 16953. MIF(m) in panel 4 is between 7117 and 3767. The maximum to minimum ratio for both MIF(m) is about 1.89.

Line 346: „ .. pattern should follow the Earth's axis rotation."

The working here is not very precise: "rotation of the Earth's axis". What exactly do you mean? The orientation of the Earth's axis is fixed relative to the stars.

Response:

The sentence has been revised to "…. relative to the ecliptic plane".

Line 353: „The MIF(s) with smoke uptake on is represented by a purple line, while the MIF(s) with smoke uptake turned off is depicted by an orange line."

No, it is the other way around!

Response:

The "on" and "off" were misplaced. The sentence is now consistent with the figure.

Line 371/372: Error estimates of the daily meteoroid input should be estimated and presented here.

Response:

The error estimations of the sodium injection rate, as well as the meteoroid material input are now presented. The error is determined by calculating the standard deviation of the detrended, unsmoothed raw MIF(s). Note that the MIF(s) presented in Fig.6 is smoothed by a 15-day running average.

---

## Author Response (AR4)

The authors have included an error estimate for the daily meteoric mass influx. I don't think the method to estimate the error is really robust, but OK.

I have some additional comments that I ask the authors to consider. Some of them are comments that I raised two or three times, but they are still not properly addressed (equation 1, "scaling" of MIF(m) to MIF(s) etc.).

Dear reviewer, we would like to personally thank you for taking the time to review the manuscript carefully. Your careful examination of the text has helped us tie up many loose ends. This work has many unique features such as incorporating data assimilation in the model, no steady assumptions used, MIF value guided by Lidar observations, and comparing MIF values from different data sources. We hope that you will find the text in its current form acceptable for publication. Thank you again for making this work much better presented.

Lines 57/58: There is a recent paper by Joe She that could be cited here:

She, C.-Y., Krueger, D. A., Yan, Z.-A., Yuan, T., & Smith, A. K. (2023). Climatology, long-term trend, and solar response of Na density based on 28 years (1990–2017) of midlatitude mesopause Na lidar observation. Journal of Geophysical Research: Space Physics, 128, e2023JA031652. https://doi.org/10.1029/2023JA031652

Response:

The suggested paper has been cited.

Equation (1): The units etc. are still not correct. Your response to my last comment was:

Reponse: In our case, the units of x0 and x1 are 1/cm3. The units of a0 and b0 are 1/cm3/second.

This cannot be correct, looking the equations. If $a_0$ and $b_0$ have the same units, then the second term in the first equation is dimensionless, the first term is not!

Same problem with the second equation. Moreover, the term in the exponential function is not dimensionless. Something is wrong or missing here.

Response:

Thank you for pointing that out. We should have noticed it ourselves. Our apologies for the oversight. The units of the production (a0) and the loss rate (b0) are 1/cm³/s and 1/s, respectively.

We have added the units in the text.

"Where $x_0$ is the value of the current step. In the simulation, it is the number density of the species. $a_0$ (1/cm³/s) is the production of the species, $b_0$ (1/s) is the loss rate of the species, $\Delta t$ is the step size in time, and $x_1$ is the value of the next step. The units for $x_0$, x₁, and $c$ are $1/cm^3$."

Line 171: "all Na(2p)"

Your response to my last comment was:

Response: Na(2p) has been changed to Na(2p) in Table 2 and in the caption.

Sorry, but this is not correct, the "P" should be upper case (as I requested in my last review already). This is textbook nomenclature for excited electronic states.

Response:

The commonly used Notation is Na($^2$P$_J$). Please refer to the references below.

Hossain, Md Mosarraf, et al. "Highly varying daytime sodium airglow emissions over an equatorial station: a case study based on the measurements using a grating monochromator." *Earth, Planets and Space* 66 (2014): 1-10.

Plane, John, et al. "On the sodium D line emission in the terrestrial nightglow." *Journal of atmospheric and solar-terrestrial physics* 74 (2012): 181-188.

Plane, John MC, Wuhu Feng, and Erin CM Dawkins. "The mesosphere and metals: Chemistry and changes." *Chemical reviews* 115.10 (2015): 4497-4541.

Bag, T., M. V. Sunil Krishna, and Vir Singh. "Modeling of Na airglow emission and first results on the nocturnal variation at midlatitude." *Journal of Geophysical Research: Space Physics* 120.12 (2015): 10-945.

Koch, Julia, et al. "Comparison of mesospheric sodium profile retrievals from OSIRIS and SCIAMACHY nightglow measurements." *Atmospheric Chemistry and Physics Discussions* 2021 (2021): 1-20.

Line 229: "The seasonal column densities of both ALO and CSU profiles is similar" "is" is not correct

Response:

"is" has been changed to "are".

Line 313: "The unit of this figure is [1/unit time/cell of equal area]."

Well, if it is „a.u." then it's better to write „a.u." (arbitrary units)

Response:

Done

Figure 6 and related text: You have to mention that MIF(m) has been scaled or fit to MIF(s). You explained it to me in your last reply, which is of course fine, but the reader also has to know it. Please mention it in the text. Perhaps I overlooked it, but I read the whole paper again and could not find it.

Response:

The sentence 'MIF(m) is in arbitrary units and has been linearly scaled to match the amplitude of MIF(s).' has been added to line 348, where Figure 6 is discussed.